# Lipoprotein-biomimetic nanostructure enables efficient targeting delivery of siRNA to *Ras*-activated glioblastoma cells *via* macropinocytosis

Jia-Lin Huang[1,*], Gan Jiang[1,*], Qing-Xiang Song[1], Xiao Gu[1], Meng Hu[1], Xiao-Lin Wang[1], Hua-Hua Song[1], Le-Pei Chen[1], Ying-Ying Lin[2], Di Jiang[3], Jun Chen[3], Jun-Feng Feng[2], Yong-Ming Qiu[2], Ji-Yao Jiang[2], Xin-Guo Jiang[3], Hong-Zhuan Chen[1] & Xiao-Ling Gao[1]

Hyperactivated *Ras* regulates many oncogenic pathways in several malignant human cancers including glioblastoma and it is an attractive target for cancer therapies. *Ras* activation in cancer cells drives protein internalization via macropinocytosis as a key nutrient-gaining process. By utilizing this unique endocytosis pathway, here we create a biologically inspired nanostructure that can induce cancer cells to 'drink drugs' for targeting activating transcription factor-5 (ATF5), an overexpressed anti-apoptotic transcription factor in glioblastoma. Apolipoprotein E3-reconstituted high-density lipoprotein is used to encapsulate the siRNA-loaded calcium phosphate core and facilitate it to penetrate the blood–brain barrier, thus targeting the glioblastoma cells in a macropinocytosis-dependent manner. The nanostructure carrying ATF5 siRNA exerts remarkable RNA-interfering efficiency, increases glioblastoma cell apoptosis and inhibits tumour cell growth both *in vitro* and in xenograft tumour models. This strategy of targeting the macropinocytosis caused by *Ras* activation provides a nanoparticle-based approach for precision therapy in glioblastoma and other *Ras*-activated cancers.

[1] Department of Pharmacology, Institute of Medical Sciences, Shanghai Jiao Tong University School of Medicine, 280 South Chongqing Road, Shanghai 200025, China. [2] Department of Neurological Surgery, Renji Hospital, School of Medicine, Shanghai Jiao Tong University, 1630 Dongfang Road, Shanghai 200127, China. [3] Department of Pharmaceutics, Key Laboratory of Smart Drug Delivery, Ministry of Education & PLA, School of Pharmacy, Fudan University, 826 Zhangheng Road, Shanghai 201203, China. * These authors contributed equally to this work. Correspondence and requests for materials should be addressed to H.-Z.C. (email: hongzhuan_chen@hotmail.com) or to X.-L.G. (email: shellygao1@sjtu.edu.cn).

The Ras pathway is commonly hyperactivated in cancers, and ~30% of all human cancers harbour at least one mutation of a *Ras* gene. The discovery of frequent activation and mutations in *Ras* family members indicates that the oncogenic Ras represents an attractive target for cancer therapy. Although efforts to target Ras have been undertaken for decades[1–3], direct pharmacologic inhibition of Ras has been a major challenge as most of small molecules targeting Ras exhibiting low potency[4]. Therefore, strategies that target the remarkable steps of *Ras* activation indirectly represent attractive alternatives for efficient anticancer therapy.

Macropinocytosis is a highly conserved endocytic process by which extracellular fluid and its contents are internalized into cells through large, heterogeneous vesicles known as macropinosomes. It is stimulated by oncogenic *Ras* and utilized as a unique mechanism for transportation of extracellular protein into the *Ras*-activated cancer cells. Rho-family guanosine triphosphatase (GTPase), which is a subfamily of Ras superfamily, triggers the macropinocytosis progression following the stimulation of extracellular components like growth factors or other signals[5,6]. Three *Ras* family members including *KRas*, *HRas* and *NRas* are expressed in all mammalian cells, and promote oncogenesis when mutation occurs, which produce the functional redundancy of GTPase and downstream cascades such as the macropinocytosis pathway[7]. Cancer cells have metabolic dependencies that distinguish them from their normal counterparts. Among these dependencies the typical one is the increased use of the amino acid glutamine to fuel anabolic processes[8]. A recent research found that in pancreatic tumour, *Ras*-activated cancer cells use macropinocytosis as a very important pathway to transport extracellular protein into the cells to serve as the source of glutamine for anabolic metabolism[9,10]. Likewise, activated *Ras* in glioblastoma cells and lung cancer cells also induces the accumulation of macropinosomes to internalize extracellular energy[11,12]. Given the fact that the macropinocytosis pathway is highly activated in *Ras*-activated cancer cells, rather than that in normal cells, strategies targeting the macropinocytosis pathway provide novel therapeutic approaches to treat the refractory tumours such as pancreatic cancer, lung cancer and glioblastoma[13]. Therefore, we proposed the hypothesis that a protein-based natural nanostructure might enhance the *Ras*-activated cancer cells to drink, in which macropinocytosis may serve as a unique mechanism to target *Ras* activation, and thus provide an efficient and universal platform for tumour-targeting drug delivery.

Glioblastoma is one of the most infiltrating, aggressive and poorly treated brain tumours. Progress in genomics and proteomics has paved the way for identifying potential therapeutic targets for treating glioblastoma, yet the vast majority of these leading drug candidates for the treatment of glioblastoma are ineffective, mainly due to the restricted drug delivery to the tumour cells. Increased expression and high levels of Ras-guanosine triphosphate (Ras-GTP) have been demonstrated in glioblastoma cell lines and tumour resection specimens in comparison with normal brain and lower grade astrocytomas[14,15]. Therefore, here we utilized glioblastoma as a tumour model to investigate the potential of tumour-targeting drug delivery strategy based on *Ras* activation-associated macropinocytosis.

Lipoproteins, natural nanoparticles, play a biological role and are highly suitable as a platform for delivering imaging and therapeutic agents. By mimicking the endogenous shape and structure of lipoproteins, lipoprotein-inspired nanoparticles can remain in circulation for an extended period of time, while largely evading the mononuclear phagocyte system in the body's defenses. In particular, high-density lipoprotein (HDL), the

smallest lipoprotein, is of interest, because of its ultra-small size and favourable surface properties. Our recent work has constructed apolipoprotein E3-reconstituted high-density lipoprotein (ApoE-rHDL) as an efficient nanoplatform that possesses blood–brain barrier (BBB) permeability for the therapy of Alzheimer's disease[16]. Very interestingly, we found that the cellular uptake of ApoE-rHDL in glioblastoma cells is much higher than that in normal primary astrocytes. In addition, the cellular uptake of ApoE-rHDL in glioblastoma cells was largely inhibited by the inhibitors of macropinocytosis, amiloride and ethylisopropylamiloride (EIPA), indicating that macropinocytosis might serve as a unique mechanism for the glioblastoma-specific accumulation of ApoE-rHDL.

To justify the concept of utilizing the enhanced macropinocytosis pathway as an efficient strategy for targeting drug delivery to the *Ras*-activated cancer cells, here we applied ApoE-rHDL as the drug delivery nanocarrier and glioblastoma as the disease model. Recent advances in RNA interference (RNAi)-based gene silencing coupled with accumulating knowledge about the role of specific genes in cancers have opened the door for the development of novel classes of precision therapeutics. However, delivery of intact, functional small interfering RNA (siRNA) into the cytoplasm of targeted cells especially in glioblastoma cells *in vivo* remains challengeable. For evaluating the potential of ApoE-rHDL as a nanoplatform for tumour-targeting siRNA delivery, activating transcription factor-5 (ATF5), an over-expressed anti-apoptotic transcription factor in glioblastoma[17,18], was chosen as the target. To enable high siRNA loading and efficient lysosome escape, siRNA entrapped by calcium phosphate (CaP) nanoparticles was introduced as a solid core of ApoE-rHDL[19]. The resulting ApoE-rHDL with a CaP core was named as CaP-rHDL. The Ras and macropinocytosis-dependent cellular uptake of CaP-rHDL and its ability to enable efficient intracellular delivery of siRNA were investigated in both glioblastoma cell lines and patient-derived glioblastoma initiating cells (GICs). The *in vivo* tumour-targeting efficiency of CaP-rHDL and the macropinocytosis dependence were also determined in intracranial C6 and GICs glioblastoma-bearing mice models. After that, the anti-glioblastoma activity of ATF5 siRNA-loaded CaP-rHDL (ATF5-CaP-rHDL) was evaluated both *in vitro* and *in vivo*.

Overall, by applying this nanoplatform, we efficiently deliver ATF5 siRNA to *Ras*-activated brain cancer cells, where the nanoparticle is uptaken by macropinocytosis in a Ras-dependent mechanism.

## Results

**Preparation and characterizations of siRNA-loaded CaP-rHDL.** siRNA-loaded CaP-rHDL (siRNA-CaP-rHDL) was prepared through the procedure as shown in Fig. 1a with that loaded with scrambled negative control siRNA (NC-siRNA) and ATF5 siRNA named as NC-CaP-rHDL and ATF5-CaP-rHDL, respectively. First, a biodegradable nano-sized CaP was applied to condense siRNA using the water-in-oil micro-emulsion method described by Li *et al.*[19]. Then, an amphiphilic phospholipid, dioleoylphosphatydic acid (DOPA), which is known to strongly interact with cations at the interface, was applied as the inner leaflet lipid to coat on the surface of the CaP cores via electrostatic interaction[20]. Subsequently, a neutral lipid, 1,2-dimyristoyl-*sn*-glycero-3-phosphocholine (DMPC), was added as the outer leaflet lipid to encapsulate the CaP core with the resulting lipid nanocarrier named siRNA-CaP-LNC. Finally, siRNA-CaP-rHDL was prepared by incubating siRNA-CaP-LNC with apolipoprotein E3 (ApoE3), which was previously confirmed to be responsible for binding to low-density lipoprotein receptor

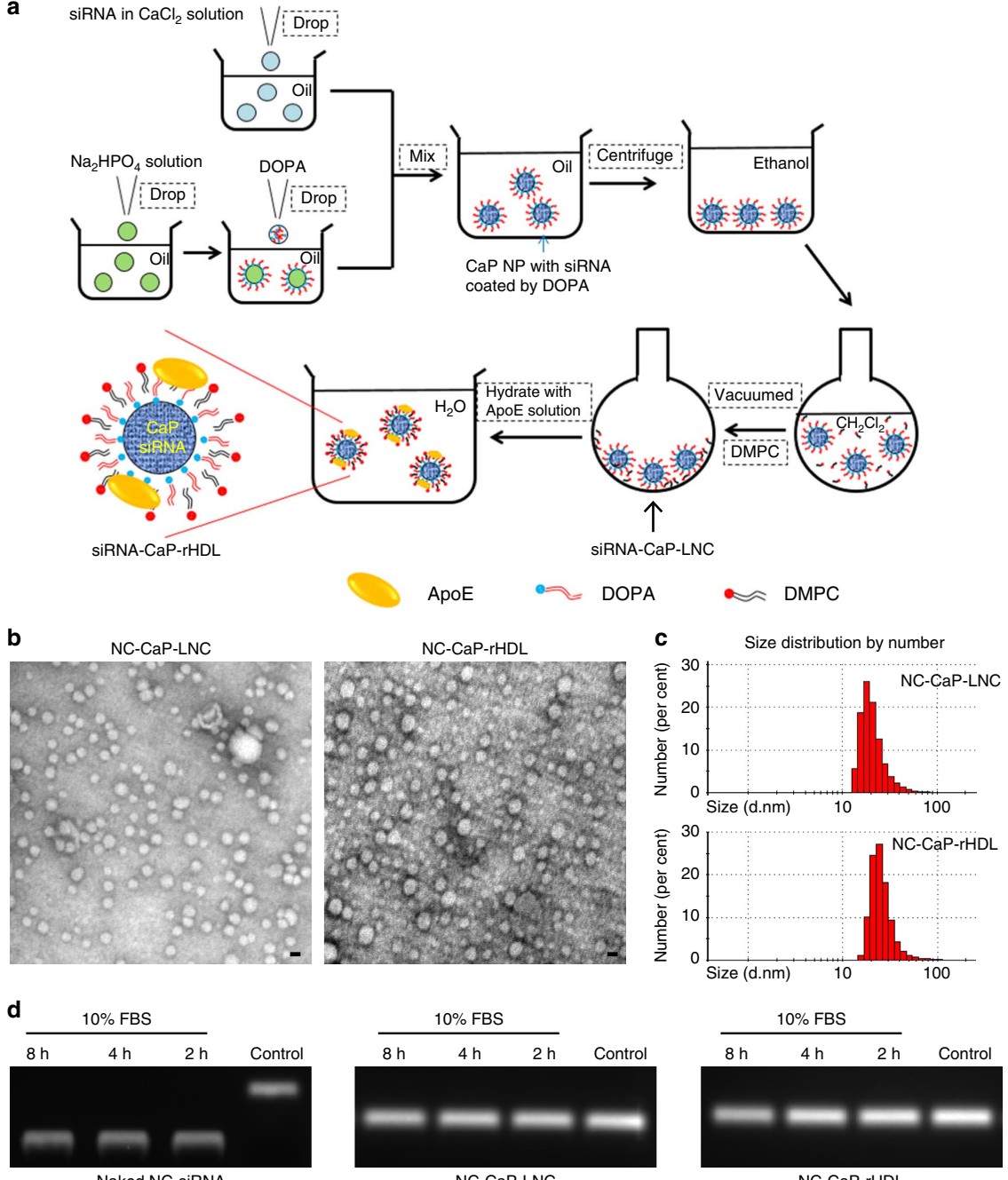

**Figure 1 | Preparation and characterization of siRNA-loaded CaP-rHDL (siRNA-CaP-rHDL).** (**a**) The outline for the preparation of siRNA-CaP-rHDL. (**b**) Morphology and particle size of negative control siRNA-loaded CaP-LNC (NC-CaP-LNC) and CaP-rHDL (NC-CaP-rHDL) under a transmission electron microscope after negative staining with sodium phosphotungstate solution (1.75%, w/v). Scale bar, 20 nm. (**c**) Particle size distribution of NC-CaP-LNC and NC-CaP-rHDL analysed by dynamic light scattering via a Zetasizer. (**d**) Serum stability of siRNA loaded by NC-CaP-LNC and NC-CaP-rHDL. Naked NC-siRNA, NC-CaP-LNC and NC-CaP-rHDL were dissolved in PBS containing 10% serum and incubated at 37 °C. At different incubation times, aliquots containing 1 μg siRNA was collected, destroyed by lysis buffer to release siRNA and subjected to siRNA detection via agarose gel electrophoresis.

(LDLR) and low-density lipoprotein receptor-related protein 1 (LRP1) that are expressed in the BBB, blood–brain tumour barrier (BBTB) and glioblastoma cells[21–25].

1,1′-dioctadecyl-3,3,3′,3′-tetramethylindocarbocyanine perchlorate (DiI) or 1,1′-dioctadecyl-3,3,3′,3′-tetramethylindotricarbocyanine Iodide (DiR), a long-chain dialkylcarbocyanine lipophilic tracer widely used for liposome and cell plasma membrane labelling, was incorporated into the membrane of CaP-loaded lipid nanocarrier (CaP-LNC) or CaP-rHDL for fluorescent

labelling. Dynamic light scattering (DLS) analysis showed that the particle sizes of siRNA-loaded or fluorescent-labelled CaP-rHDL and CaP-LNC were all in the range of 20–40 nm. The polydispersity index (PDI) of all the nanoparticles was between 0.2 and 0.4, indicating a low-to-moderate size distribution (Supplementary Table 1). The zeta potential of siRNA-loaded or fluorescent-labelled CaP-rHDL was more negative than that of the corresponding CaP-LNC, suggesting the efficient incorporation of ApoE3, a negative-charged protein,

into the lipid membrane. Moreover, from Supplementary Table 1, we can see that fluorescent labelling or siRNA loading did not change either the size or the zeta potential of both CaP-LNC and CaP-rHDL. Under transmission electronic microscope (TEM), NC-CaP-rHDL exhibited compact and spherical morphology (Fig. 1b). Most of them were close to 20 nm in diameters, which were in reasonable agreement with the results of DLS measurements (Fig. 1c).

Using FAM-labelled negative control siRNA (FAM-siRNA) as the indicator, the encapsulation efficiency of siRNA in CaP-rHDL (FAM-CaP-rHDL) and CaP-LNC (FAM-CaP-LNC) were found to be approximately 50% at the loading capacity of about 1% (w/w), which was comparable to previous reports described by Li et al.,[19] but higher than that of other lipoprotein-based nanoparticles[26]. We also study the stability of siRNA carried by CaP-LNC and CaP-rHDL, finding that in the presence of 10% serum, NC-siRNA loaded in both CaP-rHDL and CaP-LNC remained stable after 8-h incubation (Fig. 1d), while naked NC-siRNA totally degraded after only 2-h incubation. This evidence confirmed the efficient encapsulation of siRNA in CaP-LNC and CaP-rHDL and its ability to protect siRNA from degradation in physiological conditions.

**Macropinocytosis-mediated cellular uptake of CaP-rHDL.** Efficient cellular uptake is the premise for anticancer siRNA delivery. Cellular uptake of CaP-rHDL in a glioblastoma cell line was evaluated qualitatively via fluorescent microscopy analysis using DiI as the fluorescent probe. As shown in Fig. 2a, the cellular uptake of CaP-rHDL was much higher than that of CaP-LNC. Quantitative analysis revealed that the cellular uptake of CaP-rHDL was concentration, incubation time and temperature dependent, in which CaP-rHDL at the DMPC concentration 5 μg ml$^{-1}$ for 3-h incubation at 37 °C showed 110-fold greater uptake than CaP-LNC (Fig. 2b,c). This evidence suggests that the incorporation of ApoE into CaP-LNC plays an important role in facilitating the cellular uptake of the nanocarrier. More interestingly, it was found that the uptake of CaP-rHDL in rat primary-cultured astrocytes (Supplementary Fig. 1) was lower than that in C6 glioblastoma cells with the ratio one to seven (Fig. 2d), indicating that the ApoE-incorporated CaP-rHDL might be used as a nanocarrier for glioblastoma cells-targeting drug delivery.

It has been reported that LDLR and LRP1, the receptors of ApoE3, were highly expressed in glioblastoma cells[27–29], therefore, the receptor-mediated cellular internalization could be a major mechanism for the enhanced cellular uptake of CaP-rHDL. To test this hypothesis, the cellular uptake of CaP-rHDL in C6 glioblastoma cells was evaluated in the presence of excessive ApoE3. To our surprise, it was found that the cellular uptake of CaP-rHDL in C6 glioblastoma cells was stimulated by excessive but relatively low concentrations of ApoE (3.125–100 μg ml$^{-1}$), and remarkably inhibited at extremely high concentration of ApoE3 (2 mg ml$^{-1}$) (Fig. 3a). As receptor-mediated endocytosis is generally inhibited in the presence of its specific ligand, this evidence suggests that beside the ApoE receptor-medicated cellular uptake, other pathways are also likely involved in the cellular internalization of CaP-rHDL. To reveal the mechanism, an endocytosis inhibition experiment was performed and the cellular uptake of CaP-rHDL was evaluated in C6 cells in the presence of different endocytosis inhibitors including filipin, chlorpromazine, amiloride, EIPA, colchicine, cytochalasin D, nocodazole, monensin, brefeldin A and NaN$_3$ + deoxyglucose. The cellular uptake of CaP-rHDL was inhibited by almost 80% in the presence of amiloride and EIPA, both of which inhibit macropinosome formation without affecting other endocytic pathways[30]. In addition treatment with colchicine (microtubules

depolymerization agent), cytochalasin D (actin-disrupting agent), nocodazole (microtubules depolymerization agent), monensin (lysosome inhibitor), brefeldin A (Golgi apparatus destroyer) and NaN$_3$ + deoxyglucose (energy-depletion agent) decreased the internalization of CaP-rHDL by 30.1, 33.7, 40.2, 23.3, 18.7 and 46.3%, respectively (Fig. 3b). In contrast, neither caveolae nor clathrin-mediated endocytosis inhibitors (filipin and chlorpromazine, respectively) affected the cellular uptake ($P > 0.05$). This finding indicates that macropinocytosis is the major pathway for the internalization of CaP-rHDL in C6 glioblastoma cells. Consistent with these data, confocal microscopy analysis showed that DiI-CaP-rHDL (red) was co-localized (yellow) with the specific macropinocytosis marker FITC-70kDa dextran (green) with the colocalization coefficient of 82.5% (Fig. 3c). In addition, we used TEM to visualize the cellular uptake of CaP-rHDL as the golden standard to confirm this pathway at the ultrastructural level. It is known that macropinosomes are heterogeneous in size (0.2–5 μm in diameter), while clathrin-coated vesicles and caveolae are usually more homogeneous, spherical in shape and much smaller (85–150 nm diameter)[31]. As shown in Fig. 3d, CaP-rHDL appeared to be engulfed as a cluster in the vacuoles of various sizes formed by extension and bending of nearby lamellipodia that appeared at the initial stage of macropinosome formation. In contrast, the numbers of these membrane-enclosed organelles or spacious pseudopod loops were much less following the treatment of CaP-LNC. The above evidence suggests that macropinocytosis is induced following the treatment with nanostructure containing ApoE3 and serves as an important mechanism for cancer cells to 'drink' the protein-based nanocarrier. Macropinocytosis, a highly conserved endocytic process by which extracellular fluid and its contents are internalized into cells, is stimulated by activated Ras and utilized as a unique mechanism for nutrient uptake in Ras-activated cancer cells. Here, to determine if the macropinocytosis-mediated cellular uptake of CaP-rHDL is Ras-dependent, we employed the cell lines including human glioblastoma cell lines U87 and U251, human pancreatic adenocarcinoma cell line MIA PaCa-2 (KRas G12C mutation) and colorectal cancer cell line SW-480 (KRas G12V mutation), all showed higher Ras activity than astrocytes, a pancreastic cell line BxPC-3 and a colon cell line Caco-2, which express wild-type KRas[32–35], for the analysis. Along with their higher Ras activity, those KRas-mutated tumour cell lines showed higher cellular association of DiI-CaP-rHDL, compared with the KRas wild-type control cell lines (Fig. 4a,b). A straightforward linear relation was achieved between the cellular uptake of CaP-rHDL and the intracellular Ras-GTP level ($R^2 = 0.8687$) (Fig. 4a–c). We further used lentivirus-mediated anti-Ras short hairpin RNA (shRNA) to knockdown general all Ras expression in U87, U251, MIA PaCa-2 and SW-480 cells (Supplementary Fig. 2), finding that blockade of cellular Ras activity reduced the cellular association of DiI-CaP-rHDL (Fig. 4d). Such effect was not Ras-subtype dependent as knocking down the expression of KRas, HRas or NRas all reduced cellular Ras activity (Supplementary Fig. 3) and significantly inhibited the cellular uptake of both the marker of macropinocytosis tetramethylrhodamine-labelled high-molecular-mass dextran (TMR-dextran) and DiI-CaP-rHDL (Fig. 4e). The evidence strongly suggests that the Ras activation-dependent macropinocytosis serves as a unique mechanism for targeting delivery of CaP-rHDL to the Ras-activated cancer cells.

To determine whether the cancer cell-targeting delivery strategy exploited earlier can be translated to a human system, we further investigated the cellular uptake of CaP-rHDL and its macropinocytosis dependence in a patient-derived glioblastoma model. As glioblastoma is composed of various subclones of heterogeneous cells, we utilized patient-derived glioblastoma

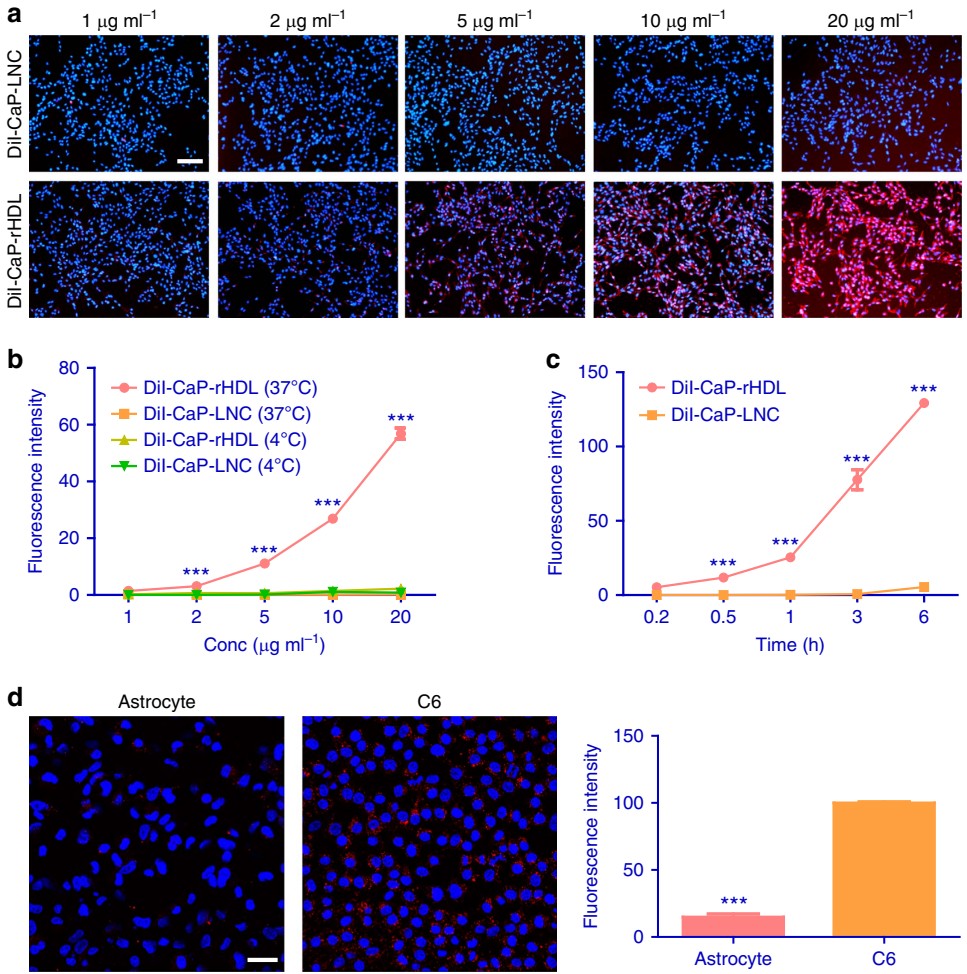

**Figure 2 | Cellular uptake of CaP-rHDL in an active and cell type-specific manner.** (**a**) Cellular uptake of DiI-labelled CaP-LNC (DiI-CaP-LNC) and CaP-rHDL (DiI-CaP-rHDL) after incubation for 3 h at the DMPC concentration ranged from 1 to 20 µg ml$^{-1}$ in C6 glioblastoma cells at 37 °C. Red: DiI-labelled nanoparticles. Blue: nuclei. Scale bar, 200 µm. (**b**) Quantitative cellular uptake of DiI-CaP-LNC and DiI-CaP-rHDL in C6 glioblastoma cells at 4 °C and 37 °C, respectively, after incubation for 3 h at the nanoparticle concentrations from 1 to 20 µg ml$^{-1}$. (**c**) Cellular uptake of DiI-CaP-LNC and DiI-CaP-rHDL in C6 glioblastoma cells after different incubation time ranged from 0.2 to 6 h at the nanoparticles concentration of 5 µg ml$^{-1}$ at 37 °C. For **b** and **c**, data represent mean ± s.d. (n = 3). ***P < 0.001 significantly higher than the cellular uptake of CaP-LNC at 37 °C. The significance of the differences was evaluated by two-tailed Student's t-test. (**d**) Cellular uptake of DiI-CaP-rHDL in C6 glioblastoma cells and primary astrocytes after incubation for 1.5 h at the nanoparticles concentration of 5 µg ml$^{-1}$. Scale bar, 50 µm. Data represent mean ± s.d. (n = 3). The significance of the differences (***P < 0.001) was evaluated by two-tailed Student's t-test.

initiating cells (GICs) (Supplementary Fig. 4) which closely mirror the phenotype and genotype of primary tumours for the evaluation[36]. It was reported that in glioblastoma, the wild-type *Ras* is hyperactivated, and the high expression of Ras combined with Akt mediates glioblastoma formation[37–39]. In addition, overexpression of Ras can also commit transformation of the neural stem cells to GICs in a mouse model[40]. We thus reasoned that the *Ras* activation-related macropinocytosis pathway may also play a crucial role in the nutrient gaining in GICs and in the uptake of CaP-rHDL. To test this hypothesis, we first performed western blot to evaluate Ras expression in the GICs derived from two patients, finding that the levels of Ras protein in the GICs were even higher than that in C6 cells (Fig. 5a). We then determined the GICs uptake of CaP-rHDL and its Ras and macropinocytosis dependence. As expected, efficient and time-dependent accumulation of CaP-rHDL in the GICs were observed under a live-cell imaging system (Supplementary Movie 1). Laser confocal imaging also showed that CaP-rHDL distributed deep into the GICs (Supplementary Movie 2), and that EIPA significantly inhibited the internalization progress in both

samples (Fig. 5b). In line with these findings, flow cytometry analysis confirmed that more than 60% of the cells were associated with DiI-CaP-rHDL after 3-h incubation, and that the pretreatment with EIPA resulted in a significant decrease in the cellular uptake of DiI-CaP-rHDL (Fig. 5c). Knocking down the expression of general all Ras in GICs also blocked cellular association of DiI-CaP-rHDL, linearly correlating with the decline of the intracellular Ras-GTP levels (Fig. 5d,e). These results supported our hypothesis that cellular uptake of CaP-rHDL depends on Ras-mediated macropinocytosis in GICs.

***In vivo* tumour-targeting efficiency of CaP-rHDL.** To determine the tumour-targeting delivery efficiency of CaP-rHDL *in vivo*, real-time near infrared (NIR) fluorescent imaging was performed using a near infrared fluorescent probe DiR to label CaP-rHDL for minimizing the animal autofluorescence background and rendering deeper tissue penetration. As shown in Fig. 6a, compared with CaP-LNC, CaP-rHDL exhibited higher fluorescent intensity of the nanoparticles in C6 cells-transplanted

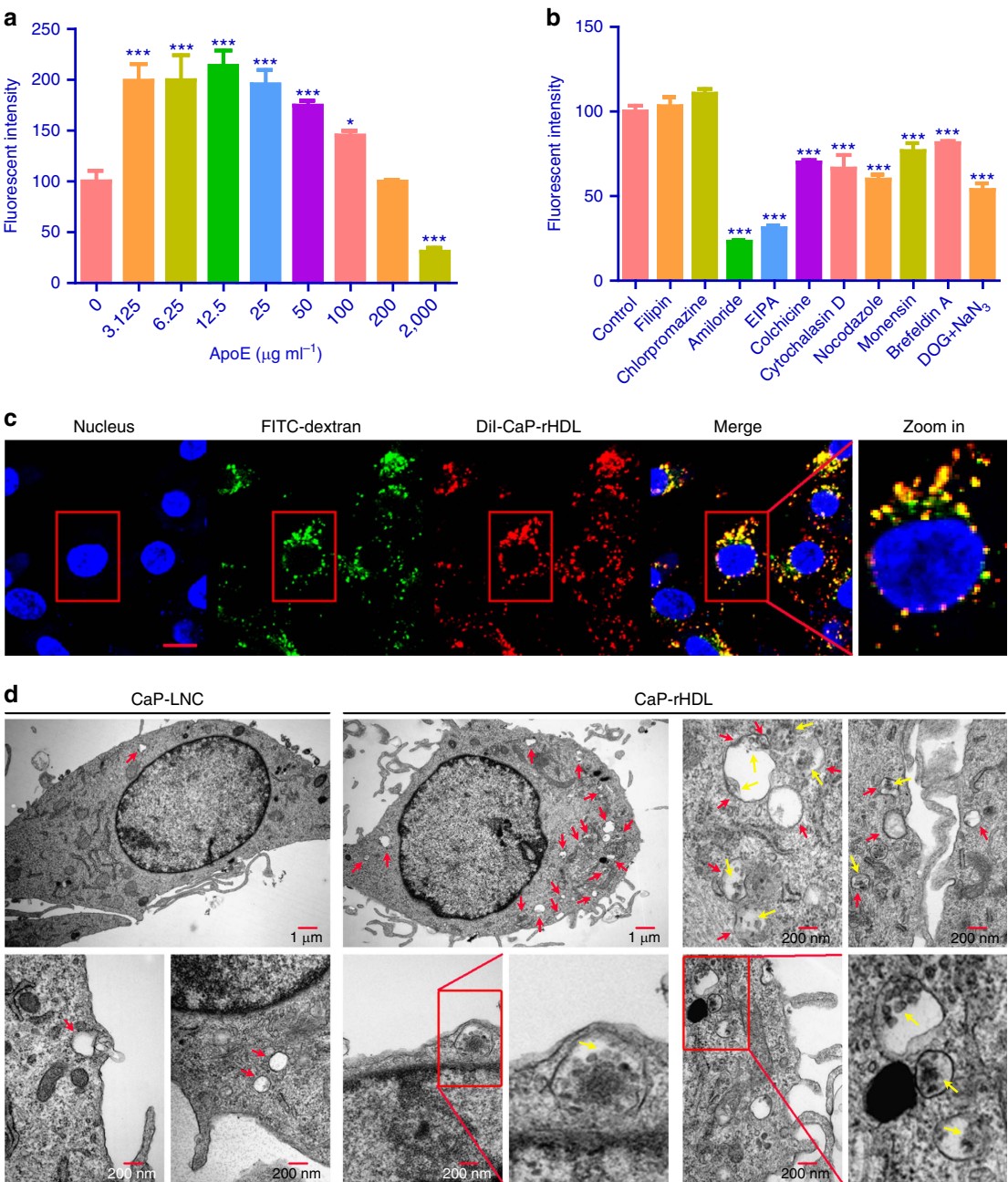

**Figure 3 | Cellular uptake of CaP-rHDL in C6 glioblastoma cells via the macropinocytosis pathway.** (**a**) Cellular association of DiI-CaP-rHDL in the presence of ApoE3 at the concentrations ranged from $0\,\mu g\,ml^{-1}$ to $2\,mg\,ml^{-1}$ in C6 glioblastoma cells. (**b**) Cellular association of DiI-CaP-rHDL in the presence of different endocytosis inhibitors. For **a** and **b**, data represent mean ± s.d. ($n = 3$). *$P < 0.05$, ***$P < 0.001$ significantly different with that of the non-treated control. Two-tailed Student's $t$-test was used for statistical analysis. (**c**) Colocalization of DiI-CaP-rHDL to macropinocytosis marker FITC-70kDa dextran. Scale bar, 10 μm. (**d**) Macropinocytosis-mediated internalization of CaP-rHDL observed under TEM. C6 glioblastoma cells were incubated with CaP-LNC or CaP-rHDL for 3 h, fixed and processed for observation under thin-section electron microscopy. Red arrows indicated macropinosomes. Yellow arrows indicated CaP-rHDL internalized via macropinocytosis.

glioblastoma xenografts from 4 to 24 h following injection. This result was confirmed by imaging the isolated organs at 24 h post-injection (Fig. 6b). Moreover, to intuitively evaluate the BBB permeability and targeting efficiency of CaP-rHDL, the frozen sections of brains from the C6 glioblastoma-bearing mice treated with DiI-CaP-rHDL or DiI-CaP-LNC were observed under a laser scanning confocal microscopy at 3 h post-injection. As shown in Supplementary Fig. 5a, CaP-rHDL was found to efficiently penetrate across the BBB while CaP-LNC showed poor permeation. Tumour section analysis of fluorescence displayed a minimal level of CaP-LNC at the central tumour site; in contrast,

higher accumulation of CaP-rHDL fluorescence was observed in the tumour (Fig. 6e). To determine if this tumour-specific accumulation of CaP-rHDL is also macropinocytosis dependent, EIPA, an inhibitor of macropinocytosis, was intraperitoneally (i.p.) injected to the nude mice bearing intracranial C6 glioblastoma[41]. Such EIPA pretreatment was found to lead to notable inhibition of tumour accumulation of CaP-rHDL (Fig. 6c–e).

The capacity of CaP-rHDL in glioblastoma-targeting delivery of siRNA was further verified in non-obese diabetic/severe combined immunodeficient (NOD/SCID) mice model bearing patient-derived intracranial glioblastoma using Cy5-labelled

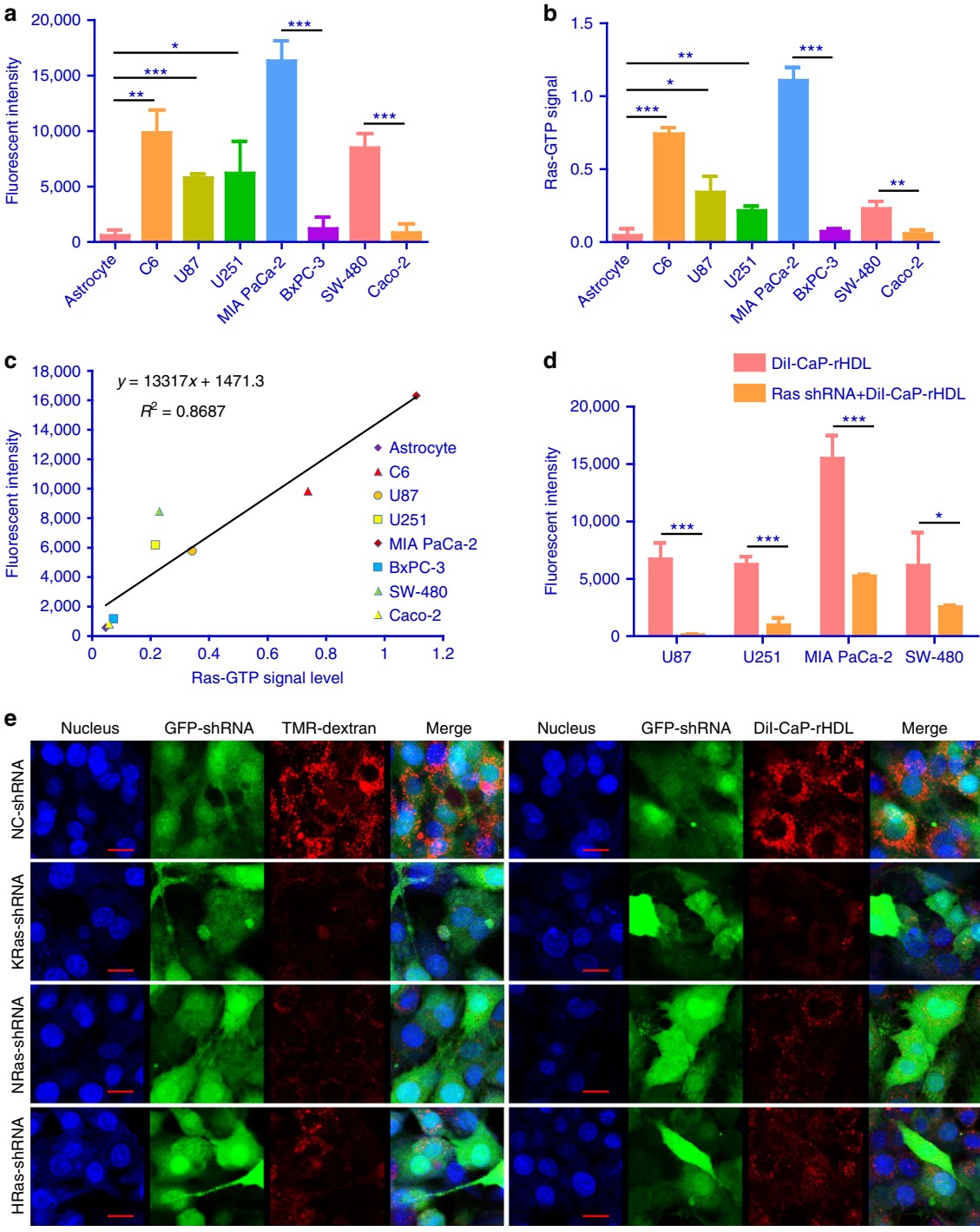

**Figure 4 | Macropinocytosis-mediated cellular uptake of CaP-rHDL in a Ras activation-dependent manner.** (**a**) Quantitative analysis of the cellular uptake of DiI-CaP-rHDL in astrocytes, C6, U87, U251, MIA PaCa-2, BxPC-3, SW-480 and Caco-2 cells after incubation at 37 °C for 3 h at the DMPC concentration of 5 μg ml$^{-1}$. (**b**) The level of Ras-GTP in astrocytes, C6, U87, U251, MIA PaCa-2, BxPC-3, SW-480 and Caco-2 cells evaluated via a Ras activation assay kit. (**c**) A linear regression was established to model the relationship between the cellular uptake of CaP-rHDL and the intracellular Ras-GTP level in the different cell types ($R^2 = 0.8687$). (**d**) Knockdown of general all Ras led to a reduction in the cellular uptake of DiI-CaP-rHDL. Ras knockdown was achieved following the application of a shRNA expression lentivirus system. (**e**) Knockdown of each Ras-subtype led to a reduction in the cellular uptake of both the marker of macropinocytosis TMR-dextran and DiI-CaP-rHDL. Ras knockdown was achieved following the application of shRNA expression lentivirus systems. A comparable level of transfection efficiency between the various shRNAs was indicated by monitoring the intensity of green fluorescent protein (GFP), which is concomitantly expressed with each shRNA. Compared with that in those cells transfected with a negative control shRNA (NC-shRNA), the cellular uptake of TMR-dextran (red) and DiI-CaP-rHDL (red) were reduced in cells transfected with KRas-specific shRNA, NRas-specific shRNA and HRas-specific shRNA. Scale bar, 10 μm. For **a,b** and **d**, data represent mean ± s.d. ($n = 3$). The significance of the differences between two groups (*$P < 0.05$, **$P < 0.01$, ***$P < 0.001$) was evaluated by two-tailed Student's $t$-test.

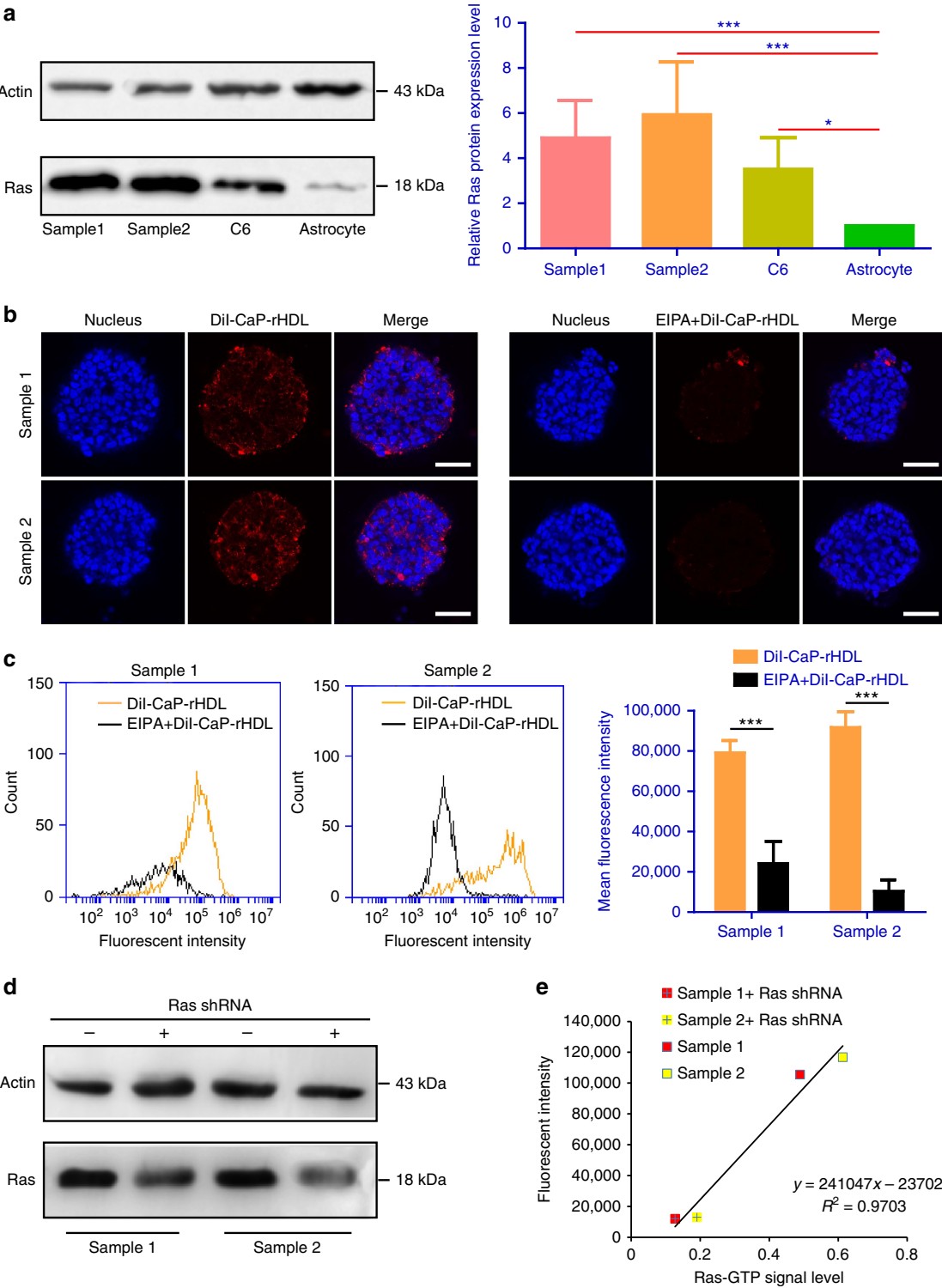

**Figure 5 | Patient-derived GICs captured CaP-rHDL in a Ras activity and macropinocytosis-dependent manner.** (**a**) Ras-protein level in the GICs derived from two patients (samples 1 and 2), C6 glioblastoma cells and normal astrocytes as determined by a western blot analysis. (**b**) Qualitative and (**c**) quantitative analysis of the cellular uptake of DiI-CaP-rHDL in the GICs derived from two patients after incubation at 37 °C for 3 h at the concentration of 20 μg ml$^{-1}$ in the absence/presence of EIPA (75 μM). Red: DiI-labelled nanoparticles. Blue: nuclei. Scale bar, 200 μm. (**d**) Western blot analysis confirmed the lower Ras-protein level in GICs after knocking down general all Ras via a shRNA expression lentivirus system. (**e**) A linear regression was established to model the relationship between the cellular uptake of CaP-rHDL and the intracellular Ras-GTP level in GICs ($R^2 = 0.9703$). Data represent mean ± s.d. ($n = 3$). The significance of the differences between two groups (*$P < 0.05$, ***$P < 0.001$) was evaluated by two-tailed Student's $t$-test.

negative control siRNA (Cy5-siRNA) as the cargo and indicator. Cy5-siRNA-loaded CaP-LNC which contains 1,2-distearoryl-sn-glycero-3-phosphoethanolamine-N-poly(ethylene glycol) 2000 (DSPE-PEG-2000) (Cy5-CaP-LNC-PEG) and is similar with the nanoparticles which have shown good *in vivo* behaviour and RNAi effect[19], was here used as the control formulation.

The characterization of Cy5-CaP-LNC-PEG was shown in Supplementary Table 1 and Supplementary Fig. 6. Twenty days after intracranial implantation, despite the highly heterogeneous glioblastoma formed in the brain of NOD/SCID mice, along with a better BBB permeation, Cy5-siRNA-loaded CaP-rHDL (Cy5-CaP-rHDL) was found to achieve higher accumulation and deeper penetration at the tumour site than Cy5-CaP-LNC-PEG (Fig. 6f–h; Supplementary Fig. 5b). In concert with earlier observation, EIPA also effectively inhibited the accumulation of Cy5-CaP-rHDL at the tumour site (Fig. 6h). Altogether, these

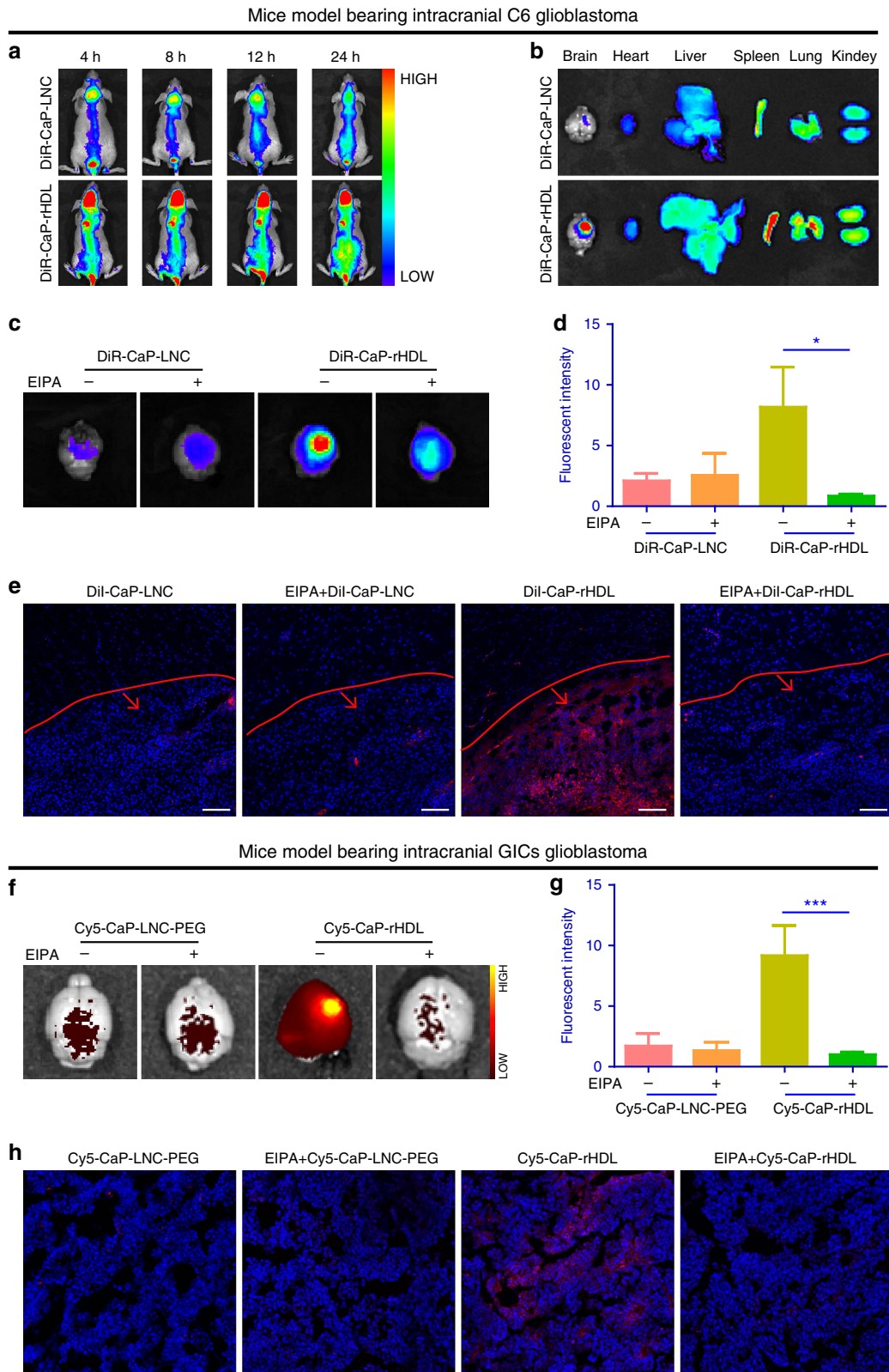

results suggest that macropinocytosis pathway contributes largely to the *Ras*-activated glioblastoma-specific accumulation of CaP-rHDL *in vivo* and that CaP-rHDL serves as a powerful nanocarrier for glioblastoma-targeting siRNA delivery.

**CaP-rHDL-mediated efficient delivery of siRNA into C6 cells.** It is well known that efficient delivery of siRNA to tumours *in vivo* requires not only high tumour accumulation, but also efficient transportation into the cytoplasm of targeted cells where siRNA recognizes and binds to RNA to silence gene function. Endosomal escape is one of the major bottlenecks for non-viral siRNA delivery, since siRNA trapped in endosomes is typically trafficked into lysosomes where siRNA is degraded[42]. In the case of macropinocytosis-mediated cellular internalization, content of macropinosomes is either degraded at the late endosome/lysosome or recycled back to the plasma membrane[31]. Herein, to determine if siRNA can escape from late-endosomes/lysosomes timely, we study the colocalization between FAM-siRNA and LysoTracker Red which is the indicator of late-endosomes and lysosomes. As shown in Fig. 7a, with the increase of the incubation time, more and more fluorescence of FAM-siRNA was found to leave lysosome and spread into the cytoplasm of C6 cells, demonstrating the ability of CaP-rHDL to release siRNA that is trapped in the late-endosomes/lysosomes.

In addition, the dissociation of siRNA from its carrier is another key step for siRNAs to exert their function. To demonstrate the deassembly process of the siRNA nanocarrier, we developed a dual cargo-loaded particle, where FAM-siRNA was loaded in the core of CaP-rHDL while DiI embedded in the surface lipid membrane. Intracellular distribution of FAM-siRNA after incubation for 2–12 h was determined and analysed under a confocal microscope. As shown in Fig. 7b, dissociation between FAM-siRNA and DiI-labelled lipid membrane was found to be a time-dependent process with the colocalization coefficient between FAM-siRNA and DiI decreased from 0.71 at 2 h to 0.46 at 12 h, suggesting that siRNA-loaded CaP-rHDL could self-disassemble following cellular internalization. Importantly, after incubated for 12 h, in certain cells, the green fluorescence of FAM-siRNA was found well diffuse throughout the cytoplasm. Collectively, the time-dependent imaging study suggested the high efficiency of CaP-rHDL in cytosol siRNA release.

**Anti-glioblastoma activity of ATF5-CaP-rHDL.** Given the earlier finding of the high cellular uptake efficiency and the outstanding endosome escape ability, we speculate that ATF5-CaP-rHDL can be delivered to tumour cells to specifically block *ATF5* gene expression, thus providing anti-tumour efficacy. ATF5 is an anti-apoptotic factor, which is highly expressed in malignant glioblastoma but not in normal brain tissues[43], and plays a key role in promoting cell survival in a variety of tumour cells[44]. So interference with its function will selectively kill neoplastic cells without affecting the viability of normal cells. In C6 glioma cells, following the treatment with ATF5-CaP-rHDL for 48 h, the

expression of ATF5 mRNA and protein was reduced by 75–85%. In contrast, ATF5 siRNA-loaded CaP-LNC (ATF5-CaP-LNC) did not have significant effects (Fig. 8a,b). We next examined the anti-proliferation effect of ATF5-CaP-rHDL in C6 cells, GICs and astrocytes using the cell counting kit-8 (CCK8) method. As shown in Fig. 8c and Supplementary Fig. 7, after incubation for 48 h, ATF5-CaP-rHDL exhibited high toxicity ($IC_{50} = 60$ nM) in C6 cells but negligible effects in astrocytes. ATF5-CaP-rHDL treatment at the dose of 100 nM for 48 h led to severe apoptosis and nuclear hyperchromatism in C6 cells[45,46] (Fig. 8d; Supplementary Fig. 8). The result was consistent with that of the quantitative assay by flow cytometry which confirmed the ability of different formulations to induce cell apoptosis followed the order: ATF5-CaP-rHDL > ATF5-CaP-LNC > NC-CaP-rHDL with the percentage of early apoptosis in cells treated with ATF5-CaP-rHDL ($53.2 \pm 10.5$%) much higher than that of those treated with ATF5-CaP-LNC ($11.1 \pm 1.3$%) and NC-CaP-rHDL ($7.1 \pm 0.15$%) (Supplementary Fig. 9a). In GICs model, ATF5-CaP-rHDL displayed the highest cell toxicity as well ($IC_{50} = 180$ nM) and led to severe apoptosis of GICs with a percentage of 70% at the siRNA concentration 200 nM (Fig. 8e,f; Supplementary Fig. 9b). Collectively, the above evidence confirmed the ability of CaP-rHDL in enabling efficient siRNA-mediated therapy *in vitro*. Finally, we evaluated the potential of CaP-rHDL as a siRNA delivery vehicle for the treatment of glioblastoma *in vivo*. In mice model raised by C6 glioblastoma cells, ATF5-CaP-rHDL, ATF5-CaP-LNC and NC-CaP-rHDL were given intravenously (i.v.) at the siRNA dose of $0.36$ mg kg$^{-1}$ on day 6, 8, 10 and 12 after surgery. One day following the last injection, the mice were killed and the tumour-loaded brains were subsequently stained for apoptotic markers. As shown in Fig. 9a and Supplementary Fig. 10a, ATF5-CaP-rHDL induced 54% cell apoptosis, whereas ATF5-CaP-LNC only promoted 6% cell apoptosis, and no apoptosis was observed in the NC-CaP-rHDL group. Such anti-tumour efficacy of ATF5-CaP-rHDL was further confirmed in the patient-derived xenograft mice model. As shown in Fig. 9b, compared with that in saline-treated animals, the expression of ATF5 at the centre of tumour site was inhibited by 80% following the ATF5-CaP-rHDL treatment. Consequently, the highest extent of apoptosis at the tumour site was induced by ATF5-CaP-rHDL with a percentage of 43%, but almost none in mice treated with NC-CaP-rHDL and ATF5-CaP-LNC (Fig. 9c; Supplementary Fig. 10b). Consistent with the apoptotic results, in C6 glioblastoma animal, survival in Kaplan–Meier survival curve showed that on day 26, the survival rate of the saline, NC-CaP-rHDL, ATF5-CaP-LNC and ATF5-CaP-rHDL-treated animals was 0, 11, 48 and 100%, respectively (Fig. 9d). In addition, the mean survival of those mice administered with saline, NC-CaP-rHDL, ATF5-CaP-LNC and ATF5-CaP-rHDL were 18, 21, 26 and 37 days, respectively. Noticeably, 33% of mice receiving ATF5-CaP-rHDL were able to survive more than 100 days. Likewise, in the GICs glioblastoma models (Fig. 9e), the mean survival of those animals administered

**Figure 6 | *In vivo* tumour-targeting efficiency of CaP-rHDL and its macropinocytosis dependence.** (**a**) *In vivo* real-time NIR fluorescent imaging of the mice bearing intracranial C6 glioblastoma at 4, 8, 12 and 24 h after i.v. injected with DiR-labelled CaP-LNC (DiR-CaP-LNC) and CaP-rHDL (DiR-CaP-rHDL). (**b**) Distribution of DiR-CaP-LNC and DiR-CaP-rHDL at 24 h post-injection in the organs of the mice bearing intracranial C6 glioblastoma. (**c**) Qualitative and (**d**) semi-quantitative analysis of the tumour accumulation of DiR-CaP-LNC and DiR-CaP-rHDL at 3 h post-injection in the mice bearing intracranial C6 glioblastoma after the pretreatment with EIPA (30 μg g$^{-1}$ body weight, i.p.) or vehicle only. (**e**) Brain distribution of DiI-CaP-LNC and DiI-CaP-rHDL at 3 h post-injection in the mice bearing intracranial C6 glioblastoma after the pretreatment with EIPA (30 μg g$^{-1}$ body weight, i.p.) or vehicle only. Red lines showed the boundary between glioblastoma and normal brain tissue. Arrows indicate the glioblastoma zones. Scale bar, 100 μm. (**f**) Qualitative and (**g**) semi-quantitative analysis of the tumour accumulation of Cy5-CaP-LNC-PEG and Cy5-CaP-rHDL at 3 h post-injection in the mice bearing GICs-derived glioblastoma after the pretreatment with EIPA (30 μg g$^{-1}$ body weight, i.p.) or vehicle only. (**h**) Brain distribution of Cy5-CaP-LNC-PEG and Cy5-CaP-rHDL at 3 h post-injection in the mice bearing GICs-derived glioblastoma after the pretreatment with EIPA (30 μg g$^{-1}$ body weight, i.p.) or vehicle only. Scale bar, 100 μm. For **d** and **g**, data represent mean ± s.d. ($n = 3$). The significance of the differences (*$P < 0.05$, ***$P < 0.001$) was evaluated by two-tailed Student's *t*-test.

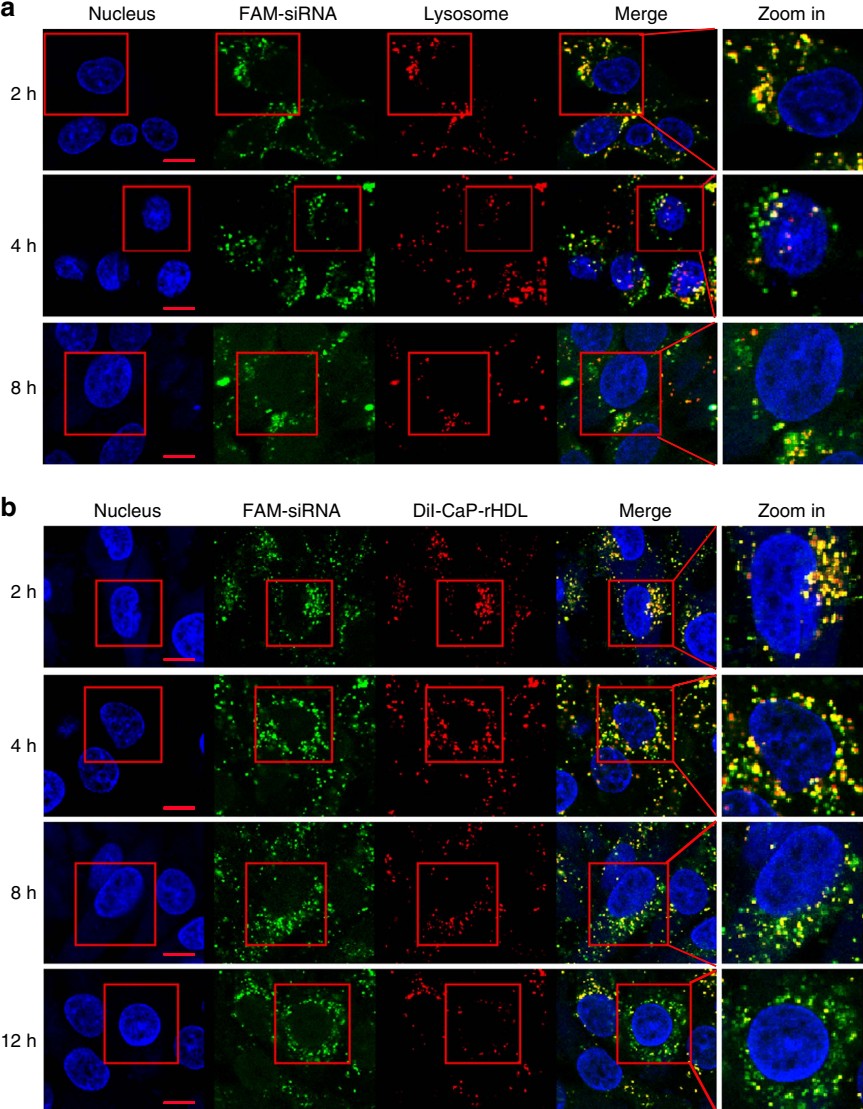

**Figure 7 | CaP-rHDL-mediated efficient delivery of siRNA in C6 glioblastoma cells.** (**a**) Lysosome escape of FAM-siRNA (green) loaded by CaP-rHDL after incubation for 2, 4 and 8 h at the siRNA concentration of 100 nM in C6 glioblastoma cells. Lysosome was indicated by LysoTracker Red. Nucleus was stained by Hoechst33342 (blue). (**b**) Dissociation of FAM-siRNA (green) from its carrier CaP-rHDL after incubation for 2, 4, 8 and 12 h at the siRNA concentration of 100 nM in C6 glioblastoma cells. DiI (red) was inserted in the lipid membrane of CaP-rHDL as the fluorescent probe. Nucleus was stained by Hoechst33342 (Blue). Scale bar: 10 μm.

with saline, NC-CaP-rHDL, ATF5-CaP-LNC and ATF5-CaP-rHDL were 25, 26, 25 and 40 days, respectively. Collectively, these *in vivo* antitumor studies provided evidence to support our hypothesis that CaP-rHDL not only efficiently delivers siRNA to the target tissue, but also facilitates the gene knockdown activity to exert its antitumor efficacy. A drug should not only be therapeutically effective, but also exhibit acceptably low toxicity. To evaluate whether ATF5-CaP-rHDL induce any adverse effect during treatment, tumour-bearing mice received the same ATF5-CaP-rHDL treatment were also subjected to behavioural, physical and biochemical effect assessment paralleled with administration with saline controls. During the treatment regimen, neither abnormalities in behaviour nor significant changes in body weight were observed following the treatment of ATF5-CaP-rHDL (Supplementary Fig. 11a). After the treatment, we also performed a pathologic examination of the major organs (heart, liver, spleen, lung and kidney) by hematoxylin and eosin staining. As shown in Supplementary Fig. 11b, no significant difference in histology was found between the ATF5-CaP-rHDL group and the saline control. The long term toxicity of ATF5-CaP-rHDL was further evaluated in normal mice following injection every two days for eight times. After the treatment, blood chemistry test and morphological observation were performed with no significant changes detected between the mice administered with saline and ATF5-CaP-rHDL (Fig. 10). These results collectively indicated the safety of ATF5-CaP-rHDL and the attractiveness of lipoprotein-biomimetic nanocarrier for *Ras*-activated tumour cell-targeting delivery of siRNA.

## Discussion
Cancer cells have metabolic dependencies that distinguish them from their normal counterparts. As the most common type of cancer cells, *Ras*-activated cancer cells use macropinocytosis as an important route for nutrient uptake to transport extracellular protein into the cells, where proteolysis occurs and yields amino acids containing glutamine that can enter the central carbon metabolism. This phenomenon indicates that macropinocytosis

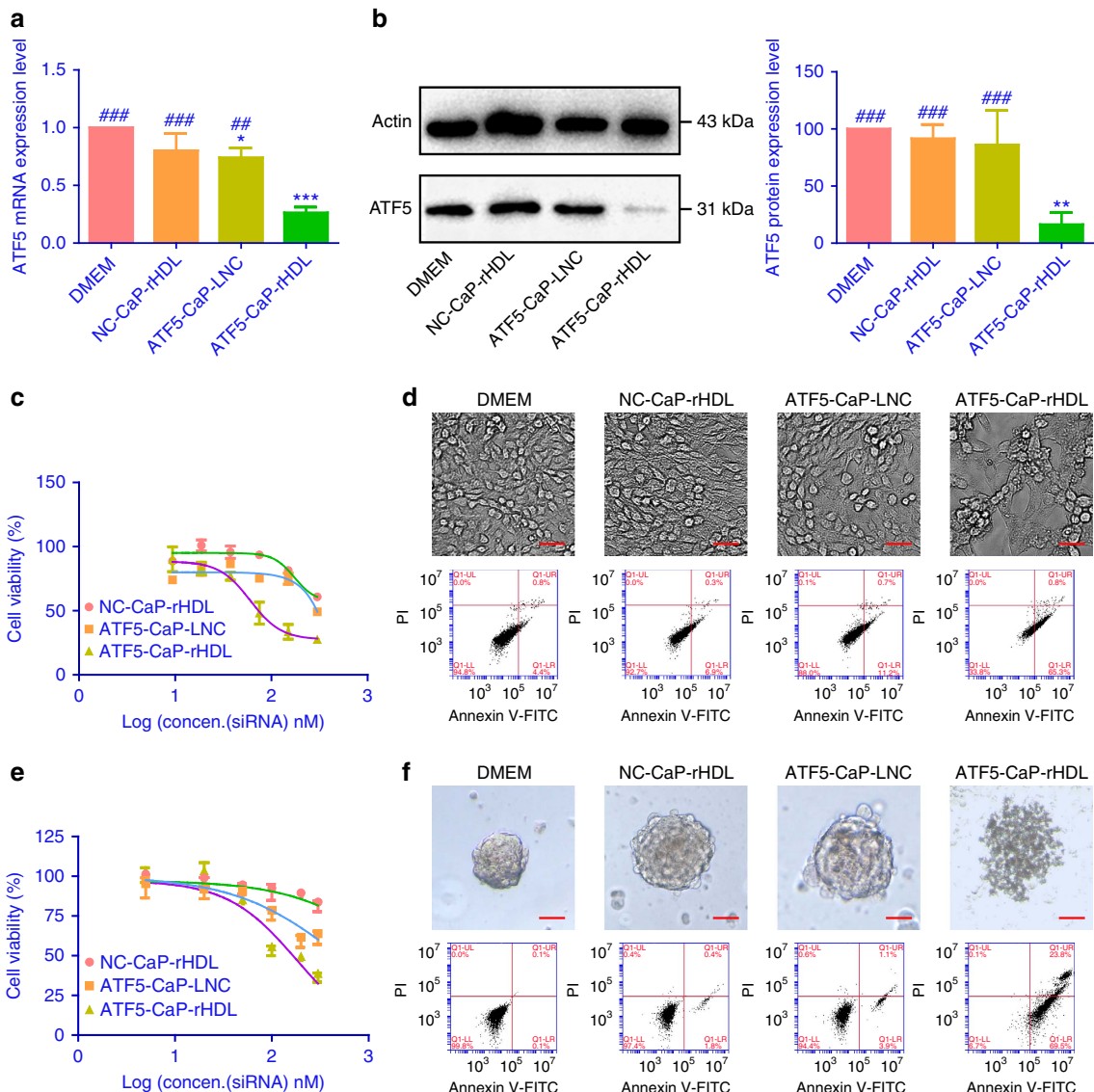

**Figure 8 | In vitro anti-glioblastoma activity of ATF5-CaP-rHDL.** (**a**) The mRNA level of ATF5 in C6 glioblastoma cells after the treatment with NC-CaP-rHDL, ATF5-CaP-LNC or ATF5-CaP-rHDL for 48 h. (**b**) The protein level of ATF5 in C6 glioblastoma cells after the treatment with NC-CaP-rHDL, ATF5-CaP-LNC or ATF5-CaP-rHDL for 48 h. For **a** and **b**, data represent mean ± s.d. ($n = 3$). The significance of the differences was evaluated by one-way ANOVA followed by Bonferroni test. $*P < 0.05$, $**P < 0.01$, $***P < 0.001$ significantly different with that of the DMEM, $##P < 0.01$, $###P < 0.001$ significantly different with that of the ATF5-CaP-rHDL. (**c**) Cell viability of C6 glioblastoma cells after the treatment with NC-CaP-rHDL, ATF5-CaP-LNC or ATF5-CaP-rHDL for 48 h. (**d**) Induction of apoptosis in C6 glioblastoma cells following 48 h incubation with various siRNA formulations at the siRNA concentrations 100 nM. The upper four images are bright field images of C6 glioblastoma cells. Scale bar, 50 μm. The bottom four images are flow cytometry results. (**e**) Cell viability of GICs after the treatment with NC-CaP-rHDL, ATF5-CaP-LNC or ATF5-CaP-rHDL for 48 h. (**f**) Induction of apoptosis in GICs following 48 h incubation with NC-CaP-rHDL, ATF5-CaP-LNC or ATF5-CaP-rHDL at the siRNA concentration 200 nM. The upper four images are bright field images of GICs. Scale bar, 50 μm. The bottom four images are flow cytometry results.

may serve as a key mechanism by which cancer cells support their unique metabolic needs, and also points to the possible exploitation of this process for the design of anticancer therapies. We were thus particularly interested in harnessing the potential of macropinocytosis to design nanoparticle-based drug-delivery system (DDS) which can target the special metabolic needs for *Ras*-activated cancer cells[47]. A biologically inspired nanostructure CaP-rHDL, derived from ApoE-rHDL that can sufficiently cross the BBB, was core-loaded with siRNA with a high efficiency and served as an effective delivery tool for anti-glioblastoma therapy. As expected, the uptake of CaP-rHDL is significantly elevated in multiple *Ras*-activated cell lines including C6 glioblastoma cells, patient-derived GICs, human glioblastoma cell lines U87 and

U251, pancreatic cancer cells MIA PaCa-2 and colorectal cell lines SW-480 cells, and the efficiency of these cellular uptakes of CaP-rHDL is linear dependent on the Ras-GTP activation level. Therefore, our findings demonstrate that Ras-dependent macropinocytosis serves as the main pathway to facilitate the cellular uptake of CaP-rHDL. The nanoparticles platform *in vivo* can also overcome the obstacle mainly attributed by the BBB to reach to tumour site and restrict tumour progression. These findings are extremely interesting as the Ras pathway is commonly deregulated in cancers, whereas the oncogenic Ras currently remains to be one of the most 'undruggable' targets.

Since both Ras deregulation and macropinocytosis occur frequently in *Ras*-activated cancer cells, CaP-rHDL thus provides a

universal nanocarrier for tumour-targeting drug delivery. In addition, distinct in many ways from the better characterized endocytosis pathways such as clathrin and caveolae-mediated endocytosis, the relative large size (0.2–5 μm), a low level of lysosomal degradation and a rapid release of molecules to the cytosols make macropinocytosis a more efficient route for intracellular delivery[48–50].

Utilizing RNA interference as an innovative therapeutic strategy has an immense likelihood to generate novel concepts

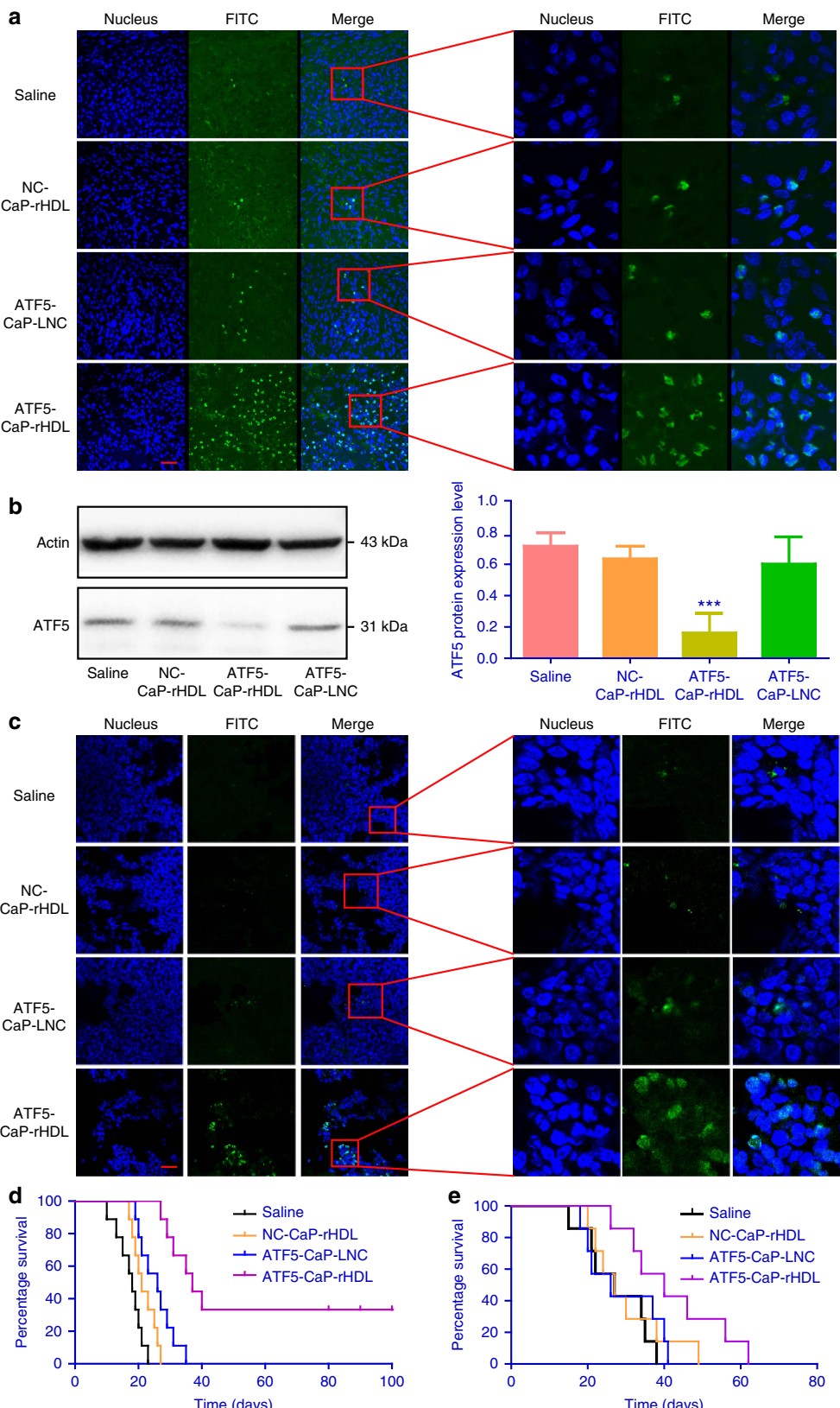

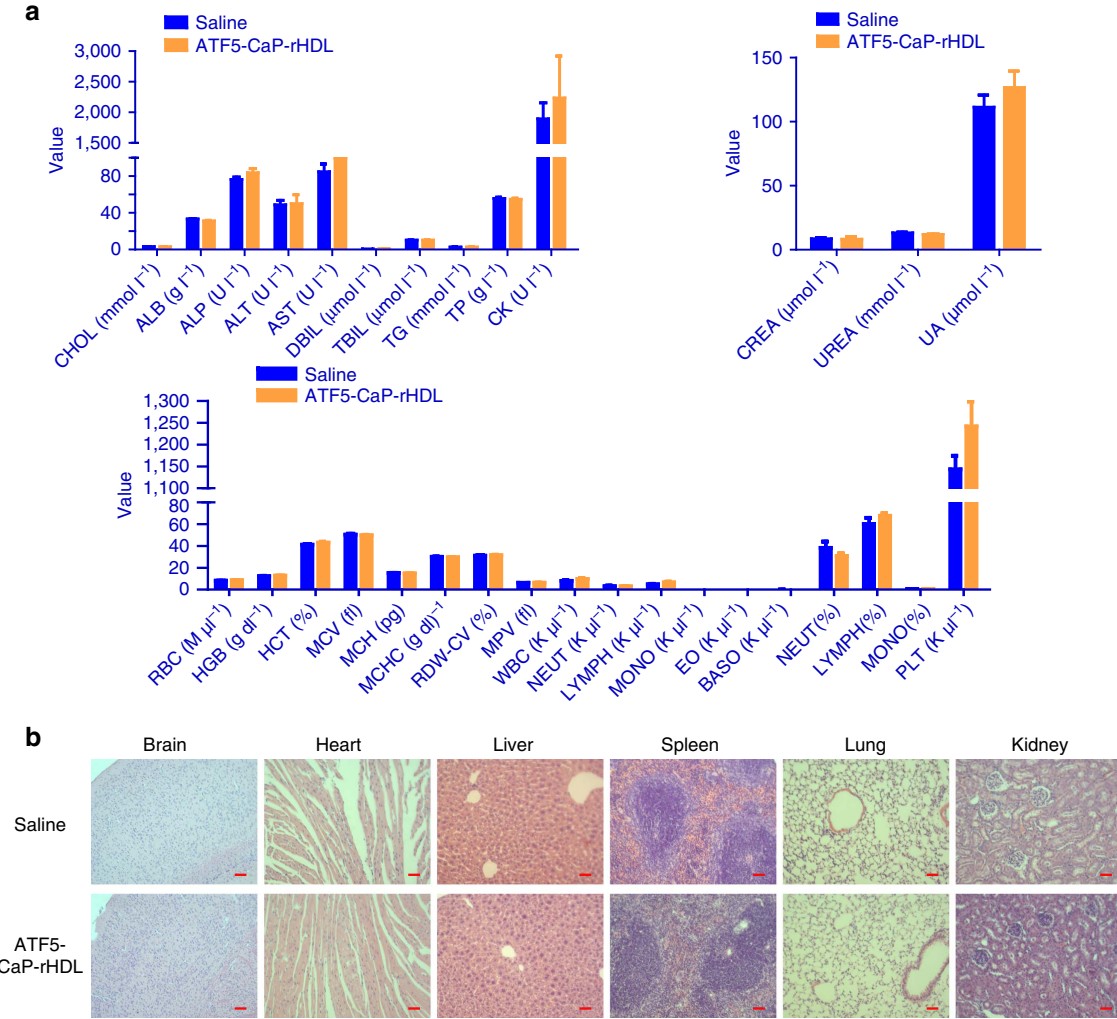

**Figure 10 | Evaluation of the safety of ATF5-CaP-rHDL on healthy ICR mice.** Healthy ICR mice were i.v. administered with saline or ATF5-CaP-rHDL at the siRNA dose of 0.36 mg kg$^{-1}$ every 2 days for a total eight doses ($n=5$). (**a**) Blood biochemistry and haematology tests were performed at one day after the last injection. ALB, albumin; ALP, alkaline phosphatase; ALT, alanine aminotransferase; AST, aspartate transaminase; BASO, basophils; CHOL, cholesterol; CK, creatine kinase; CREA, creatinine; DBIL, direct bilirubin; EO, eosinophils; HCT, haematocrit; HGB, haemoglobin; LYMPH, lymphocytes; MCH, mean corpuscular haemoglobin; MCHC, mean corpuscular haemoglobin concentration; MCV, erythrocyte mean corpuscular volume; MONO, monocytes; MPV, mean platelet volume; NEUT, neutrophils; PLT, platelet.; RBC, red blood cell; RDW-CV, red cell distribution width (coefficient of variation); TBIL, total value bilirubin; TG, triglyceride; TP, total protein; UA, uric acid; UREA, urease; WBC, white blood cell. Data represent mean ± s.d. Unpaired student's *t*-test (two-tailed) was used for comparison between two groups, and $P<0.05$ were considered significant. (**b**) Hematoxylin and eosin staining of the major organs (heart, liver, spleen, lung, kidney). Scale bar, 50 μm.

in precision medicine. Yet, delivery of RNAi payloads such as siRNAs or microRNA mimics in a safe, tissue- and cell-specific manner remains a challenge. Herein, we developed CaP-rHDL as the DDS for siRNA delivery to glioblastoma which is especially hard to target[51]. The DDS offers several advantages including BBB permeability, tumour targeting, powerful siRNA loading capacity, lysosome escape ability, biocompatibility and being able to be manipulated for redirected targeting[52]. As the expression of LDLR and LRP1 was also found in the BBB and BBTB, therefore CaP-rHDL can pass BBB and BBTB via the ApoE receptor-

**Figure 9 | ATF5-CaP-rHDL induced apoptosis at the glioblastoma site and prolonged the survival of mice bearing intracranial glioblastoma.** (**a**) Nude mice bearing intracranial C6 glioblastoma treated with saline, NC-CaP-rHDL, ATF5-CaP-LNC or ATF5-CaP-rHDL every 2 days for four times at the siRNA dose of 0.36 mg kg$^{-1}$ ($n=3$). One day after the last injection, the animals were killed with the brains collected, sectioned and stained using a TUNEL kit. Scale bar, 50 μm. (**b**) NOD/SCID mice bearing intracranial GICs glioblastoma treated with saline, NC-CaP-rHDL, ATF5-CaP-LNC or ATF5-CaP-rHDL every 3 days for five times at the siRNA dose of 0.36 mg kg$^{-1}$ ($n=3$). One day after the last injection, the animals were killed with the brains collected. ATF5 level at the tumor regions was quantified via western blot. The significance of the differences was evaluated by one-way ANOVA followed by Bonferroni test. ***$P<0.001$, significantly different with that of the saline group. (**c**) NOD/SCID mice bearing intracranial GICs glioblastoma were treated with saline, NC-CaP-rHDL, ATF5-CaP-LNC or ATF5-CaP-rHDL every 3 days for five times at the siRNA dose of 0.36 mg kg$^{-1}$ ($n=3$). One day after the last injection, the animals were killed with the brains collected, sectioned and stained using a TUNEL kit. Scale bar, 50 μm. (**d**) Kaplan–Meier survival curve of mice bearing intracranial C6 glioblastoma treated with saline, NC-CaP-rHDL, ATF5-CaP-LNC or ATF5-CaP-rHDL every 2 days for four times at siRNA dose of 0.36 mg kg$^{-1}$ ($n=9$). (**e**) Kaplan–Meier survival curve of mice bearing patient-derived GICs glioblastoma treated with saline, NC-CaP-rHDL, ATF5-CaP-LNC or ATF5-CaP-rHDL every 3 days for five times at siRNA dose of 0.36 mg kg$^{-1}$ ($n=7$).

mediated transcytosis[53,54]. The successful self-assembly with ApoE is of vital importance for tumour targeting, because it can enable the recognition of the nanostructure by the hungry cancer cells, which absorbed nutrient by macropinocytosis. CaP-rHDL also provides an ideal nanoplatform for siRNA delivery. Compared with previous work in which siRNA was modified with lipophilic moieties and inserted into the lipid monolayer of rHDL[26], the loading capacity of siRNA in CaP-rHDL is much higher, and the siRNA encapsulated inside the core can avoid the degradation by nuclease and avoid the chemical modification which might affect the efficiency of siRNA. In addition, siRNA-loaded CaP cores were readily degradable under an intracellular endosomal environment, which would be beneficial to the release of siRNA from the nanocarrier[55]. Moreover, CaP-rHDL holds the potential to be manipulated for redirected targeting[52,56–60]. For example, the current nanocarrier containing ApoE as the functional moiety exhibited relatively high accumulation in the liver. It should be interesting that other proteins or peptides replacing ApoE can also be specifically internalized by macropinocytosis with lower affinity to the non-targeted organs.

To evaluate the siRNA delivery efficacy of CaP-rHDL, ATF5 siRNA, which can induce apoptosis in tumour cells specifically, was chosen as a model drug. ATF5-CaP-rHDL was found to dramatically reduce the expression of ATF5 mRNA and protein and efficiently induce apoptosis in glioblastoma cells. As a result, ATF5-CaP-rHDL significantly prolonged the survival of treated mice with minimal side effects. The siRNA dosage (0.36 mg kg$^{-1}$ body) used *in vivo* is relatively low compared with other delivery systems for effective gene therapy in brain tumour models[61–64], which further confirmed the superiority of lipoprotein-biomimetic nanocarrier and its unique uptake mechanism for anti-glioblastoma therapy. For other tumour model like pancreatic and colorectal cancer, this CaP-rHDL nanoplatform may also induce the tumour cells to 'drink drugs' through the macropinocytosis mechanism.

Therefore, we built an efficient DDS to target the *Ras*-activated cancer cells via the macropinocytosis-mediated mechanism, and this strategy highlights an effective feature for the design of precision therapeutics.

## Methods

**Materials.** DMPC and DOPA were obtained from Avanti Polar Lipids (Alabaster, AL, USA). Full-length ApoE3 was provided by PEPROTECH (Rocky Hill, NJ, USA). DiI was provided by Invitrogen. DiR and Hoechst 33258 were provided by Sigma-Aldrich (St Louis, MO, USA). 4,6-diamidino-2-phenylindole (DAPI) was purchased from Molecular Probes (Eugene, OR, USA). Cell counting kit-8 (CCK8) was obtained from Dojindo Laboratories (Kumamoto, Japan) and Annexin V-FITC Apoptosis Detection kit was obtained from BD PharMingen (Heidelberg, Germany). All siRNAs used in rat species model were synthesized by Genepharma (Shanghai, China). ATF5 siRNA used in rat species model consisted of the sense strand 5′-CCUGUCCCUCCAUUUCACUTT-3′ and antisense strand 5′-AGUG AAAUGGAGGGACAGGTT-3′. NC-siRNA have a scrambled sequence consisted of the sense strand 5′-UUCUCCGAACGUGUCACGUTT-3′ and antisense strand 5′-ACGUGACACGUUCGGAGAATT-3′. All siRNAs used in the human species model were synthesized by Ribobio. (Guangzhou, China). The ATF5 siRNA used in the human species model consisted of the sense strand 5′-CCCAUAUCCUAC AGGCAA AdTdT-3′ and antisense strand 5′-UUUGCCUGUAGGAUAUGG GdTdT-3′. NC-siRNA used in the human species model have a scrambled sequence consisted of the sense strand 5′-UUCUCCGAACGUGUCACG UdTdT-3′ and antisense strand 5′-ACGUGACACGUUCGGAGA AdTdT-3′. The fluorescent (FAM, Cy5)-labelled negative control siRNAs have the same scrambled sequence and the dye was labelled on the antisense strand 5′. The PCR primers to detect ATF5 (forward: 5′-GTGCCTAGGGTACAGGAGGA-3′, reverse: 5′-GCAGAGGGGAGACCTAGACA-3′) and β-actin (forward: 5′-CATC GTGGGGCCGCCCTAGGC-3′, reverse: 5′-GGGCCTCGGTGAGCAGCACA-3′) were synthesized by Sangon Biotech (Shanghai). Rabbit polyclonal to ATF5 antibody (Catalogue number: ab60126), Rabbit monoclonal to Ras antibody (Catalogue number: ab52939), Rat monoclonal to CD31 antibody (species reactivity: mouse, Catalogue number: ab7388), Rabbit polyclonal to SOX2 antibody (Catalogue number: ab97959), Rabbit polyclonal to CD31 antibody (species reactivity: human, Catalogue number: ab28364) and Rabbit monoclonal to Nestin antibody

(Catalogue number: ab105389) were obtained from Abcam (Cambridge, UK), Mouse monoclonal to glial fibrillary acidic protein (GFAP) antibody (Catalogue number: MAB360) from Millipore (Temecula, CA, USA), Rabbit polyclonal to CD133 (PROM1) antibody (Catalogue number: SAB2107606) from Sigma-Aldrich. Other chemicals were obtained from Sigma-Aldrich unless otherwise indicated.

**Cells.** MIA PaCa-2, BxPC-3, SW-480, Caco-2, U87, U251 and Rat C6 glioblastoma cells were obtained from Cell Institute of Chinese Academy of Sciences (Shanghai, China) and cultured under standard conditions using the culture medium containing DMEM, 10% FBS, 1% L-glutamine, 1% penicillin–streptomycin solution and 1% non-essential amino acids. Primary astrocytes were separated from the glial cultures using a mild trypsinization protocol described by Saura *et al.*[65]

GICs were obtained from patient-derived glioblastoma samples. Tumour samples were obtained during the surgery from a 4-year-old patient and a 6-year-old patient in the Department of Neurosurgery, Shanghai Renji Hospital (Shanghai, China). Informed consent was obtained from guardians before the surgery as approved by the ethics committee. Samples were collected after surgical resection and immersed immediately (15 min) in ice-cold Dulbecco's modified Eagle's medium (DMEM-F12) containing 10% penicillin and streptomycin. The tissue was washed with PBS, minced and digested with 0.05% Typsin-EDTA & Type 4 Collagenase. Digestion was stopped every five minutes to collect the cells via centrifugation at 2,000 r.p.m. for 5 min. The cells were then washed once with PBS and cultured in the neurosphere medium supplemented with growth factors (NMGF) at a relatively low density ($1 \sim 3 \times 10^5$ cells ml$^{-1}$) in T25 tissue culture flask under standard conditions. The NMGF solution contains DMEM/F12, B27 supplement, penicillin and streptomycin, 20 ng ml$^{-1}$ bFGF and 20 ng ml$^{-1}$ EGF[66,67].

**Animals.** The NOD/SCID mice, ICR mice and Balb/c nude mice (male, 4–5 weeks, $20 \pm 2$ g) were purchased from the Shanghai SLAC Laboratory Animal (Shanghai, China). The animals were housed in the specific pathogen-free animal facility with free access to food and water. All animal experiments were approved by the Animal Experimentation Ethics Committee of Shanghai Jiao Tong University School of Medicine.

To establish C6 glioblastoma-bearing mice model, the suspension of C6 cells (500,000 cells per 5 μl in pH 7.4 PBS) were injected into right corpus striatum of nude mice with the help of a stereotaxic apparatus. To establish GICs glioblastoma-bearing mice model, the suspension of GICs (containing 1,000 spheres per 10 μl in pH 7.4 PBS, less than 20 passages) were gently pipetted before injected into the right corpus striatum of the NOD/SCID mice. After the transplantation, the animals were cultured under standard condition for 2 weeks before the subsequent experiments.

**Preparation and characterization of the nanoparticles.** The lipid-coating CaP cores were prepared via the water-in-oil reverse micro-emulsion method as described previously[66,67]. Briefly, 300 μl of 2.5 M CaCl$_2$ and 80 μl of 2 mg ml$^{-1}$ siRNA were dispersed in 20 ml cyclohexane/Igepal CO-520 (71/29V/V) solution to form a very well dispersed water-in-oil reverse micro-emulsion. The phosphate part was prepared by adding 300 μl of 12.5 mM Na$_2$HPO$_4$ (pH $>9$) in a separate 20 ml oil phase. One-hundred microliters of DOPA (20 mg ml$^{-1}$) in chloroform was added to the phosphate phase. After mixing the above two solutions for 45 min to form micro-emulsion, 40 ml of ethanol was added and the mixture was centrifuged at 12,500 g for 20 min to remove cyclohexane and the surfactant. After being extensively washed by ethanol for three times, the pellets were dissolved in 1 ml chloroform and stored in a glass vial for further modification. For the preparation of siRNA-CaP-rHDL, 750 μl of the above mentioned CaP core was mixed with 4 mg DMPC and subjected to drying under a high vacuum for 1 h at room temperature using a Büchi Rotavapor R-200 (Büchi, Germany). The lipid film was then rehydrated in 4 ml of 0.01 M PBS buffer (pH 7.4) by vortexing intermittently for 10 min. The liposome solution (siRNA-CaP-LNC) was then stored at 4 °C after brief sonication. For the preparation of siRNA-CaP-rHDL, ApoE3 (0.8 mg) was added and further incubated with siRNA-CaP-LNC (containing 4 mg DMPC) at 37 °C for 36 h. FAM-siRNA, Cy5-siRNA, NC-siRNA and ATF5 siRNA-loaded CaP-LNC and CaP-rHDL were prepared with the same procedure only with different siRNA sequences. DiI, DiR-labelled nanoparticles were also prepared with the same procedure by adding DiI and DiR in the DMPC solution. For the preparation of Cy5-CaP-LNC-PEG, 750 μl of the above mentioned CaP core was mixed with 4 mg DMPC and 2.5 mg DSPE-PEG-2000, and then subjected to drying under a high vacuum for 1 h at room temperature and rehydrating with 4 ml PBS as described above.

The morphology and size of NC-CaP-LNC, NC-CaP-rHDL and Cy5-CaP-LNC-PEG were observed under a Hitachi H-7650 TEM (Hitachi, Japan) after negative staining with 1.75% sodium phosphotungstate solution. The particle size distribution and zeta potential were measured by photon correlation spectroscopy (Zetasizer Nano-ZS90, Malvern Instruments, UK) utilizing a 4 mW He-Ne laser operating at 633 nm and a detector angle of 90°.

To qualitatively calculate the encapsulation efficiency of siRNA in CaP-rHDL, FAM-siRNA was used as the indicator. For the analysis, FAM-CaP-rHDL was dissolved in lysis buffer (2 mM EDTA and 0.05% Trixton-100 in pH 7.8 Tris

buffer) at 65 °C for 10 min to release the entrapped siRNA before the fluorimetry measurement.

To check the stability of nanoparticles in serum, NC-CaP-LNC, NC-CaP-rHDL and naked NC-siRNA were incubated with 10% FBS at 37 °C for 2, 4 and 8 h. The samples were then analysed by electrophoresis using 2% agarose gel to monitor the release or degradation of nanoparticles. The gel was run at 120 V for 15 min and subsequently imaged via an ODYSSEY infrared imaging system.

**Quantification of cellular uptake of CaP-rHDL.** C6 cells were seeded into 96-well plates at the density of 5,000 cells per well and allowed to attach for 24 h. After that, the cells were exposed to different concentrations of DiI-CaP-rHDL and DiI-CaP-LNC. For qualitative experiment, the nanoparticles were added at the DMPC concentration ranged from 1 to 20 µg ml$^{-1}$. After incubation at 37 °C or 4 °C for 3 h, the cells were washed twice with cold PBS buffer, fixed with 4% formaldehyde for 15 min and then subjected to fluorescent microscopy analysis (Leica DMI4000 B, Germany). For quantitative analysis, after fixation, the cells were nuclei-stained with Hoechst 33258 for 15 min away from light and then subjected to analysis under a KineticScan HCS Reader (Thermo Scientific, USA). For evaluating the time related cellular association of the nanoparticles, the incubation time was ranged from 0.2 to 6 h at the DMPC concentration of 5 µg ml$^{-1}$. To compare the efficiency of cellular uptake of CaP-rHDL in primary astrocytes and C6 glioblastoma cells, the cells were both incubated with CaP-rHDL at 5 µg ml$^{-1}$ for 1.5 h.

To study the mechanism of cellular uptake of CaP-rHDL in C6 cells, the cells were pre-incubated with a serious concentrations of ApoE3 and different endocytosis inhibitors, including 10 µM filipin, 5 µM chlorpromazine, 5 mM amiloride, 25 µM EIPA, 10 µM colchicine, 3 µM cytochalasin D, 20 µM nocodazole, 200 nM monensin, 18 µM brefeldin A, 5 mM NaN$_3$ and 25 mM deoxyglucose for 0.5 h. After that, 5 µg ml$^{-1}$ DiI-CaP-rHDL was added into each well and incubated for 1 h. Then the cells were treated as described before quantitative study.

To study the Ras activity-dependent cellular uptake efficiency, astrocytes, C6, U87, U251, MIA PaCa-2, BxPC-3, SW-480 and Caco-2 cells were seeded into 24-well plates at the density of 10$^5$ cells per well and allowed to attach for 24 h. After that, the cells were exposed to 5 µg ml$^{-1}$ DiI-CaP-rHDL and incubated for 1.5 h. The cells were washed, fixed with paraformaldehyde and analysed on a flow cytometry. The cellular uptake of DiI-CaP-rHDL in GICs was also qualitative analysed by confocal microscopy and quantitative analysed by flow cytometry.

**Ras activity assay.** Endogenous Ras-GTP levels were measured using an enzyme-linked immunosorbent assay-based G-LISA kit (Cytoskeleton, Denver, CO, USA, catalogue #BK131) following the manufacturer's instructions. Briefly, cells were plated and allowed to grow to roughly 50% confluence before being washed with PBS and lysed in 100 µl of ice-cold lysis buffer in the presence of protease inhibitor cocktail. The lysate was clarified by centrifugation at 10,000 g for 1 min, and snap frozen in liquid nitrogen. After normalizing protein concentration using PrecisionRed (Cytoskeleton), samples were added in triplicate to 96-well plates coated with Ras GTP-binding protein and incubated at 4 °C for 30 min at 300 revolutions per minute (r.p.m.). After washing, bound Ras-GTP levels were determined by subsequent incubations with an anti-Ras antibody and a secondary horseradish peroxidase (HRP)-conjugated antibody, followed by addition to an HRP detection reagent. Ras activity was quantified by measuring absorbance at 490 nm. Background was determined with a negative control well loaded with lysis buffer, and experiments for each cell type were repeated three times.

**Colocalization assay under confocal microscopy.** To determine whether macropinocytosis is the key mechanism for cellular uptake of DiI-CaP-rHDL in C6 glioblastoma cells, the cells were treated with 5 µg ml$^{-1}$ DiI-CaP-rHDL at 37 °C for 1.5 h in the presence of 1 mg ml$^{-1}$ FITC-labelled high-molecular-mass dextran (FITC-dextran) which is an established marker of macropinocytosis. Then the cells were treated as described above before the colocalization assay. Images were captured under a confocal microscope (Zeiss LSM 710) and analysed via Image Pro Plus software. The total particle area per cell was determined from at least three fields that were randomly selected from different regions across the entirety of each sample.

To verify that FAM-siRNA can escape from lysosomes, CaP-rHDL loaded with FAM-siRNA at the siRNA concentration of 100 nM was incubated with C6 cells for 2, 4 and 8 h. The cellular distribution of FAM-siRNA was then determined and analysed under a confocal microscope (Zeiss LSM 710) with LysoTracker Red as the indicator of lysosome.

To illuminate that FAM-siRNA can be released from the carrier CaP-rHDL, C6 cells were exposed to FAM-siRNA and DiI double-loaded CaP-rHDL at the siRNA concentration of 100 nM for 2, 4, 8 and 12 h. The fluorescence signals in the cells were imaged using laser confocal scanning microscopy.

**TEM analysis of macropinocytosis.** To examine the ultrastructure of cellular uptake of CaP-rHDL in C6 glioblastoma cells, the cells were incubated with CaP-rHDL or CaP-LNC for 3 h at the DMPC concentration of 100 µg ml$^{-1}$. After that, the cells were washed with PBS for two times, fixed with 2.5% glutaraldehyde at 4 °C for 2 h, scratched off from the flask, centrifuged at 1,500 r.p.m. for 5 min with the pellet re-suspended in 2.5% glutaraldehyde, stored at 4 °C until post-fixation in

1% OsO$_4$ in 1 M PB, and finally subjected to ultra-thin section and microscopic analysis under a Hitachi 7600 electron microscope.

**Interference RNA-mediated knockdown of Ras.** Ras knockdown was achieved through the shRNA expression lentivirus systems provided by HanYin Biotech. (Shanghai, China). shRNA sequences of KRas shRNA, NRas shRNA and HRas shRNA used in rat species model are GGACTCCTACAGGAAACAAGT, GCAAATTAAGCGCGTGAAAGA and GCAGATCAAGCGGGTGAAAGA, respectively. shRNA sequences for general all Ras used in human species model is AATTCAAAAAACAAGAGGAGT ACAGTGCAATCTCTTGAATTG CACT GTACTCCTCTTCG. C6, U87, U251, MIA PaCa-2, SW-480 cell lines and GICs were transfected with the lentiviral particles containing Ras shRNA, and selected with 2 µg ml$^{-1}$ puromycin for 3 days. Those cells treated with DMEM were used as the controls. Efficient knockdown of Ras was confirmed by immune-blotting with Ras-specific antibodies.

**Live imaging.** Live imaging of GICs was performed using a confocal system (Zeiss) installed with a microscope at ×10 magnification (Axio Observer. Z1). Images were recorded with an AxioCamMR3 camera every 6 min to avoid toxicity. Exposure time was constant. DiI-CaP-rHDL was directly added into the dish at the DMPC concentration of 20 µg ml$^{-1}$. The collected images were converted to AVI movies by using the ZEN lite software.

**In vivo real-time imaging and glioblastoma distribution.** In C6 glioblastoma-bearing nude mice, the biodistribution of CaP-LNC and CaP-rHDL following intravenous administration was studied via a Maestro in vivo imaging system (CRi, MA, USA) at the excitation wavelength 748 nm and emission wavelength 780 nm with DiR as the fluorescent probe and via tumour section analysis using DiI as the indicator. Six mice were randomly divided into two groups and i.v. injected with the DiR/DiI-labelled nanoparticles at the dose of DMPC 20 mg kg$^{-1}$. The fluorescent images of the live animals were taken at 4, 8, 12 and 24 h after the injection. At 24 h post-injection, the tumour-bearing mice were killed with the organs harvested for ex vivo imaging. For tumour section analysis, at 3 h post-injection, the mice were anaesthetized, and heart perfused with saline and 4% paraformaldehyde. Then the brains were collected, fixed in 4% paraformaldehyde, dehydrated in 10%, 30% sucrose solution, imbedded in OCT (Sakura, Torrance, CA, USA), frozen at −80 °C and sectioned at 14 µm before observation under a confocal microscope. In GICs glioblastoma-bearing NOD/SCID mice model, Cy5-siRNA was used as the cargo and indicator of CaP-rHDL.

For BBB permeability evaluation, the above frozen brain slides were firstly blocked with 20% goat serum for 1 h at room temperature, and incubated with CD31 antibody (1:300 in PBS) overnight at 4 °C, and then with Alexa Fluor 488-conjugated monoclonal IgG (1:500) as the secondary antibody. Finally, the slides were subjected to confocal microscopy analysis (LSM710, Leica, Germany) after staining with DAPI for 10 min and rinsing with PBS. Each experimental group consisted of three animals and at least five sections per tumour tissue.

To evaluate the macropinocytosis dependency of the tumour-specific accumulation of CaP-rHDL in vivo, mice bearing intracranial C6 or GICs glioblastoma (n = 3) were injected i.p. with 30 µg g$^{-1}$ body weight of EIPA. Animals treated with vehicle only (2.5% DMSO in PBS, V/V) were used as the control. At 30 min post-injection, mice were i.v. administered with CaP-rHDL and CaP-LNC at the DMPC dose of 20 mg kg$^{-1}$. Due to the short half-life of EIPA in vivo (31.2 ± 2.5 min)[68], at 2 h after the first EIPA injection, the mice were repeatedly injected i.p. with 30 µg g$^{-1}$ body weight of EIPA or vehicle. At 3 h after the first EIPA injection, the mice were killed with the brains harvested for ex vivo imaging or tumour section analysis as described above.

**qPCR analysis.** To determine the ATF5 gene expression after the treatment of ATF5-CaP-rHDL, a qPCR study was performed. For the analysis, C6 cells (5 × 10$^4$ per well) were seeded in 6-well plates (Corning, Corning, NY, USA) and allowed to grow for 12 h, and then treated with different formulations at the concentrations of 100 nM siRNA at 37 °C for 48 h. Total cell RNA was extracted with an RNeasy Mini Kit (Qiagen, Valencia, CA, USA), and cDNAs were synthesized with a SuperScript II reverse transcriptase assay (Invitrogen, Carlsbad, CA, USA). qPCR was performed with a SYBR GreenER qPCR SuperMix Universal kit (Invitrogen). Reactions were run with a standard cycling programme: 50 °C for 2 min, 95 °C for 10 min, 40 cycles of 95 °C for 15 s, and 60 °C for 1 min, on a Lightcycler 480II real-time PCR system (Roche Diagnostics, Foster City, CA, USA).

**Western blot analysis.** To detect the ATF5 protein expression, C6 cells were seeded in a six-well plate at the density of 5 × 10$^4$ cells per well and allowed to grow for 12 h. Then the cells were treated with NC-CaP-rHDL, ATF5-CaP-LNC or ATF5-CaP-rHDL at the concentration of 100 nM siRNA with those cells incubated with siRNA-free cell culture medium as the negative control. After incubation for 48 h, the cells were washed twice with PBS and lysed in lysis buffer for 30 min. The whole-cell lysate was centrifuged at 12,000 r.p.m. for 5 min. The supernatant was transferred to a new tube for measuring the protein concentration via a BCA (bicinchoninic acid) protein assay kit. To perform immunoblotting, the cell lysates

(20 μg protein per lane) were loaded on a 10% sodium dodecyl sulfate (SDS)—polyacrylamide gel at 100 V, and then transferred to nitrocellulose membranes by using standard method. After twice 15-min rinsing in tris buffered saline with tween-20 (TBST), the membranes were blocked for 1 h in a solution of 5% powdered nonfat milk in tris buffered saline (TBT) and stained with a 1:1,000 dilution of rabbit polyclonal antibody against ATF5 at 4 °C overnight. After washing for three times, the membranes were treated with a 1:300 dilution of goat anti-rabbit secondary antibody at 37 °C for 1 h. Proteins were then visualized using high sensitive enhanced chemiluminescence reagent (Sangon Biotech, China) and subsequently imaged by ODYSSEY infrared imaging system.

To detect ATF5 protein level in the glioblastoma tissue, the central tumour tissues were collected from those mice treated with saline, NC-CaP-rHDL, ATF5-CaP-LNC and ATF5-CaP-rHDL, respectively, and the ATF5 level in the tissue homogenate was analysed as described above.

The Ras protein levels in astrocytes, C6, U87, U251, MIA PaCa-2, SW-480 and GICs in the presence/absence of different Ras shRNA were also analysed as described above. The full-size blots are shown in the Supplementary Fig. 12.

**Anti-proliferation assay.** The anti-proliferation activity of ATF5-CaP-rHDL in C6 cells or GICs was evaluated by using the CCK8 assay. Briefly, the cells were seeded in a 96-well plate at the density of 5,000 cells per well or 50 spheres per well. After 12 h, NC-CaP-rHDL, ATF5-CaP-LNC and ATF5-CaP-rHDL were added into wells at the siRNA concentration ranged from 1 to 300 nM, respectively. Forty-eight hours later, 10 μl CCK8 was added into each well and incubated for 0.5 h. After that, the plates were subjected to a microplate reader (ThermoMultiskan MK3, USA) for cell viability assay at the wavelength of 450 nm.

**Cell apoptosis assay.** For cell apoptosis assay, C6 cells ($5 \times 10^4$ cells per well) or GICs (200 spheres per well) were seeded in a six-well plate and allowed to grow for 12 h. After that, the cultured medium was substituted with different formulations including NC-CaP-rHDL, ATF5-CaP-LNC and ATF5-CaP-rHDL containing 100 nM siRNA for C6 cells and 200 nM siRNA for GICs. After incubated for 48 h, the cells were trypsinized, centrifuged at 1,000 g for 5 min and then stained with FITC Annexin V Apoptosis Detection Kit I (Becton Dickinson MedicalDevices, Shanghai, China) according to the manufacture's protocol. After that, the cells undergoing apoptosis were quantified by a FACS scan Flow Cytometer (BD PharMingen, Heidelberg, Germany). The cells treated with DMEM were used as control.

**TdT-mediated dUTP nick-end labelling assay.** To detect apoptotic cells in tumour tissues, a TdT-mediated dUTP nick-end labelling (TUNEL) assay was performed by using a DeadEnd Fluorometric TUNEL System (Promega) following the manufacturer's protocol. After four or five injections, the mice bearing C6 or GICs glioblastoma were anaesthetized, and heart perfused with saline and 4% paraformaldehyde. The brains were then collected, fixed in 4% paraformaldehyde and dehydrated in 10%, 30% sucrose solution. Afterward, the brains were imbedded in OCT (Sakura, Torrance, CA, USA), frozen at −80 °C and sectioned at 14 μm. Finally, the slides were subjected to confocal microscopy analysis (LSM710, Leica, Germany) after the TUNEL staining and quantitatively analysed via Image J software. Each experimental group consisted of three animals and at least three sections per tumour tissue.

**Survival analysis.** *In vivo* anticancer activity was evaluated in mice bearing C6 glioblastoma or patient-derived GICs. The treatment was performed at 6, 8, 10 and 12 days after surgery for the animals bearing C6 glioblastoma, and at 8, 11, 14, 17 and 19 days after surgery for the mice bearing patient-derived GICs. The mice were randomly divided into four groups ($n = 9$ for C6 model and $n = 7$ for GICs model), and treated with saline, NC-CaP-rHDL, ATF5-CaP-LNC or ATF5-CaP-rHDL (siRNA dosage 0.36 mg kg$^{-1}$), respectively, via tail vein injection. The survival of each group were recorded and analysed.

**Biosafety evaluation.** To assess the adverse effects of the nanoparticles during the treatment, nude mice bearing C6 glioblastoma were i.v. injected with NC-CaP-rHDL, ATF5-CaP-LNC or ATF5-CaP-rHDL every other day for four times with those mice administered with saline as the controls. After each injection, mice behaviours were monitored, and the body weight was measured every 2 days. One days post the fourth injection, the mice were killed and their organs (heart, liver, spleen, lung and kidney) were excised, fixed in 10% formalin, embedded in paraffin, sectioned and stained with hematoxylin and eosin for histological analysis. In addition, normal ICR mice were i.v. injected with ATF5-CaP-rHDL every other day for eight times with those mice administered with saline as the controls. One day post the eighth injection, the blood samples were collected and subjected to blood chemistry tests and the organs (brain, heart, liver, spleen, lung and kidney) were excised for histological analysis.

**Statistical analysis.** All the data were presented as mean ± s.d. Unpaired student's *t*-test (two-tailed) was used for between two-group comparison and one-way ANOVA with Bonferroni tests for multiple-group analysis. Statistical significance was defined as $P < 0.05$.

**Data availability.** The data that support the findings of this study are available from the authors upon reasonable request.

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

## Acknowledgements

This study was supported by National Key Basic Research Programme (2013CB932502), National Natural Science Foundation of China (Nos 81373351, 81573382, 81373353), grants from Shanghai Science and Technology Committee (15540723700, 13NM1400500), 'Shu Guang' project supported by Shanghai Municipal Education Commission and Shanghai Education Development Foundation (15SG14), Shanghai Talent Development Fund and funding from Shanghai Jiao Tong University. We would like to thank Prof Leaf Huang, Dr Jun Li and Dr Yuhua Wang for their help in the preparation of the CaP core.

## Author contributions

X.-L.G., J.-L.H. and G.J. designed the research; J.-L.H., G.J. carried out most experiments; J.-L.H., G.J. and X.-L.G. analysed the data; Q.-X.S. provided help for most experiments; X.G. and H.-H.S. assisted with the establishment of glioblastoma-bearing mice model. X.-L.W. and J.C. assisted with pharmacokinetic analysis; M.H. and Y.-Y.L. assisted with WB; G.J. and Y.-Y.L. derived GICs; J.-F.F., Y.-M.Q. and J.-Y.J. provided tumour samples and obtained informed consents; J.-Y.J. provided ethics committee approval; D.J. assisted with characterization of the nanoparticles; X.-L.G. supervised J.-L.H., H.-Z.C. supervised G.J., X.-G.J. and J.C. supervised D.J.; X.-L.G., J.-L.H., G.J. and H.-Z.C. wrote the paper. All authors discussed the results and commented on the manuscript.
