## [Peer Review File · Nature Communications]

Reviewers' comments:

Reviewer #1 (Remarks to the Author):

This manuscript reports the development of a biomimetic lipoprotein nanoparticle (CaP-rHDL) for the efficient delivery of siRNA as the target treatment of Ras-activated brain cancer. In this work, the authors carefully characterized the Ras-activation dependency of the macropinocytosis-mediated cellular uptake of the CaP-rHDL in both glioblastoma cell line and patient-derived glioblastoma. They further demonstrated the anti-glioma activity of ATF5 siRNA-loaded CaP-rHDL both in vitro and in vivo, resulting in remarkable RNA-interfering efficiency and increased apoptosis at a low siRNA dose (0.36mg/kg). Ras pathway plays essential role in malignant transformation and is an attractive target for cancer therapies. This strategy of targeting the macropinocytosis caused by Ras activation is highly novel and could provide powerful nanoparticle-based treatment strategy for Ras-activated cancers. In my opinion, this study is potentially very exciting and is a rather comprehensive work that could be considered for publication on Nature Communications after addressing following concerns.

1. The stability of control particle, CaP-LNC for in vivo study could contribute to the significant low uptake of CaP-LNC in tumor compared with CaP-rHDL, rather than the less targeting effect assumption made by authors (Fig 6 discussion). Therefore, serum stability and pharmacokinetic study should be conducted to guide in vivo data analysis. In addition, an alternate but better control could be the formulation described in Ref 19 since it contains both cholesterol and PEG-lipid to stabilize the particle and has shown good in vivo behavior and RNAi effect. Further, due to the stability concern of CaP-LNC, a much higher and selective accumulation of CaP-rHDL in the glioma regions in Fig.6 is not a conclusive evidence to support the statement of "ApoE-reconstituted lipoprotein could overcome BBB, BBTB and enormously accumulated at the glioma site where nutrient is lack due to tumor growth".

2. A dual dye labeled CaP-rHDL (siRNA labeled with FAM, particle's membrane labeled with DiI) was used to investigate the de-assemble ability of nanocarrier. As micropinocytosis is an entire particle internalization pathway, a time-dependent imaging of dual dyes in subcellular components is suggested to track the disassembling of nanoparticles.

3. The nanoparticle name is confusing. Authors are quite liberal with the use of term, CaP-rHDL, where it stands for siRNA loaded nanoparticle in Figure 1a but also was used for imaging studies without siRNA loading. Yet, another different name was used for the toxicity study (Fig.10) (ATF5 siRNA-loaded CaP-rHDL). Please be consistent.

4. The nanoparticles labeled with DiI or DiR on the particle membrane were used for examining the delivery mechanism by fluorescence imaging technology. It is better to label the siRNA payload with fluorescence dyes, such as FAM or NiR dye, to directly monitor siRNA delivery in vitro or in vivo.

5. Please correct some inconsistent dye labeling in Fig 6 (DiR-labeled) vs in the text (DiI-labeled). The explanation for switching DiI labeling (in vitro study) to DiR labeling (in vivo study) should also be given.

6. Some important references are missed in the manuscript including the first lipoprotein redirecting report (PNAS, 2005, 102, 17757-17762) as well as the reference to the selected ATF5-siRNA sequences.

7. Please define abbreviations when they appear in the manuscript at the first time, such as Ras-GTP, DDS, CCK-8, etc.

Reviewer #2 (Remarks to the Author):

Brain tumor delivery is a significant concern. The studies involve a potential improvement in the specificity of delivery based on a system specifically geared towards ras activated cells. The initial phases of the development are interesting, but the biologic aspects of the studies are relatively

weak and would benefit from substantial additional effort. In addition, the data are poorly presented with substantial data potentially better presented as data not shown or in the supplemental data.

Major concerns:

1. The authors make assumptions regarding ras activity based on shRNA against three Ras family members and immunoblots. This is inadequate. The measurement of ras activity, not ras protein expression, should be performed with every model.
2. The authors have performed almost all studies in C6 glioma. The rationale for this model is lacking. It only makes sense to use C6 glioma if a rat host is to be used. The use of nude athymic mice supports a more relevant focus on the human models. The models presented are poorly characterized, with no molecular information. This manuscript would benefit from a much greater number of models representing a spectrum of molecular groups, at least in vitro.
3. The glioma initiating cells (GICs) lack any kind of validation. To claim GICs, it is essential to perform in vivo and in vitro analysis. The optimal approach would involve the separation of stem-like and differentiated progeny and tumor initiation in vivo with in vitro limiting dilution studies.
4. The shRNA studies for each Ras family member is somewhat concerning. Having the same response for all the Ras members suggests a lack of specificity in the effects. It would be useful to document the Ras activity after shRNA treatment. Similar studies should be performed in normal cells.
5. The in vivo outcome is interesting but has only a single, suboptimal model with a single replicate. The preclinical efficacy is the most important issue. There should be a greater effort to show that these results can be generalized. Also, the tumors should be analyzed over the long term to demonstrate delivery of the siRNA and effects on the cells. The data in Figure 9 is biased in favor of response because it is too early and intensive.
6. The sphere responses are not properly designed. The images are not useful. The proper studies are in vitro limiting dilution studies with single cells.
7. The toxicity data are not well designed. First, the single astrocyte line is not well characterized and the inclusion of these cells in a greater number of experiments for both entry and toxicity would be important. Second, in the in vivo studies, the imaging suggests that delivery to some organs may be greater than the tumor. The weight analysis and organ analysis is performed at a very early time point. The longer-term effects should be considered.
8. The ability to deliver across the blood-brain barrier was not tested and the targeting of

The reagents are exciting but the entire evaluation is not optimal and are not at the level for a high impact journal, yet.

Reviewer #3 (Remarks to the Author):

Summary: In this manuscript the authors provide evidence that ApoE-rHDL nanoparticles containing a calcium phosphate (CaP) core can be taken up effectively by glioma cells in culture (monolayer and spheroid) and in intracranial glioma in mice. Furthermore, they demonstrate that such nanoparticles can be employed to deliver siRNA targeting a pro-survival protein, ATF5, thereby inhibiting intracranial tumor growth and prolonging survival of tumor-bearing mice. The authors claim two other novel aspects of their work: 1) The mechanism of uptake of the CaP-rHDL nanoparticles entails their incorporation into macropinosomes. 2) Macropinocytosis is Ras-dependent, suggesting that the NPs may be particularly useful for targeting "Ras-activated glioblastoma".

General Comments: While the use of rHDL nanoparticles for delivery of therapeutic cargo is not entirely new, their successful use for targeted siRNA delivery to glioblastoma is potentially very important. Overall the aspects of the current studies demonstrating selective uptake of the CaP-

rHDL particles and effective ATF5 suppression by siRNA in vitro are well designed and executed. The evidence that macropinocytosis plays a key role in the process is also convincing. The endogenous Ras and Rac pathways are known to play an essential role in the control of macropinocytosis in both normal and transformed cells, so it is not surprising that knockdown of Ras expression results in reduced uptake of CaP-rHDL nanoparticles. However, the claim that that such nanoparticles represent a specific therapeutic approach for selective targeting of tumors driven by activated Ras is confusing, and the studies fall short of demonstrating this. The problem is, in part, related to the authors' use of the terms "Ras-activated" glioblastoma and "Ras-transformed cell line". Activating mutations in Ras are rare in human GBM, and the majority of the studies in this report were performed with the C6 rat glioma cell line, which does not harbor Ras mutations and thus is not a "Ras-activated cell line". It is true that expression of endogenous Ras protein is relatively high in these cells compared with normal astrocytes but this is not the same as a "Ras-activated" tumor where Ras mutation (e.g., V12) results in constitutive activation of Ras pathways and is a driver of tumorigenesis and tumor progression.

Specific Comments:

- 1) Measurements of total endogenous Ras protein expression levels (Fig. 5a) do not necessarily indicate that Ras is "hyperactivated". Determination of the percentage of activated Ras (e.g., by G-LISA Ras activation assays) would be required to draw this conclusion.
- 2) The conclusion that CaP-rHDL nanoparticles "penetrated deep and distributed extensive(ly) in the C6 glioblastoma spheroids" (pg 16) seems inconsistent with the confocal images in Supplementary Fig. 2, which show most of the DiI-labeled CaP-rHDL concentrated around the periphery of the spheroids. Regarding the same point, in Fig. 5b, the images of patient-derived spheroids appear to be whole mounts, so it is not clear whether the fluorescent cells are mostly on the surface or the interior.
- 3) The small sizes of some of the illustrations, particularly in the complex figures, make it impossible to see if the conclusions are supported. For example, in Fig. 8d, the authors conclude that treatment with ATF5 siRNA "led to hyperchromatism of the cell nuclei". That panel should be moved to the supplementary data and expanded so that nuclear morphology can be easily assessed.
- 4) In Fig. 8e, we are shown one spheroid for each condition evidence that anti-ATF5-CaP-rHDL nanoparticles inhibit growth/viability of C6 spheroids. This conclusion needs to be supported by quantitative data such as average spheroid diameter or volume, with an appropriate n and statistical analysis.
- 5) The argument that the increased apoptosis (Fig. 9) and the improved survival (Fig. 10) of mice treated with anti-ATF5 nanoparticles is specifically related to suppression of ATF5 would be strengthened by IF or WB results demonstrating that ATF5 is suppressed in the tumor tissue.
- 6) The paper contains numerous grammatical and typographical errors throughout.

Below is a detailed description of our revision according to the review comments.

Journal: *Nature Communications* (NCOMMS-16-01632)

Title: “Lipoprotein-Biomimetic Nanostructure Enables Efficient Targeting Delivery of siRNA to Ras-Activated Glioblastoma Cells via Macropinocytosis”

Review comments:

Reviewer: 1

Comments:

This manuscript reports the development of a biomimetic lipoprotein nanoparticle (CaP-rHDL) for the efficient delivery of siRNA as the target treatment of Ras-activated brain cancer. In this work, the authors carefully characterized the Ras-activation dependency of the macropinocytosis-mediated cellular uptake of the CaP-rHDL in both glioblastoma cell line and patient-derived glioblastoma. They further demonstrated the anti-glioma activity of ATF5 siRNA-loaded CaP-rHDL both in vitro and in vivo, resulting in remarkable RNA-interfering efficiency and increased apoptosis at a low siRNA dose (0.36mg/kg). Ras pathway plays essential role in malignant transformation and is an attractive target for cancer therapies. This strategy of targeting the macropinocytosis caused by Ras activation is highly novel and could provide powerful nanoparticle-based treatment strategy for Ras-activated cancers. In my opinion, this study is potentially very exciting and is a rather comprehensive work

that could be considered for publication on Nature Communications after addressing following concerns.

1. The stability of control particle, CaP-LNC for in vivo study could contribute to the significant low uptake of CaP-LNC in tumor compared with CaP-rHDL, rather than the less targeting effect assumption made by authors (Fig 6 discussion). Therefore, serum stability and pharmacokinetic study should be conducted to guide in vivo data analysis. In addition, an alternate but better control could be the formulation described in Ref 19 since it contains both cholesterol and PEG-lipid to stabilize the particle and has shown good in vivo behavior and RNAi effect. Further, due to the stability concern of CaP-LNC, a much higher and selective accumulation of CaP-rHDL in the glioma regions in Fig.6 is not a conclusive evidence to support the statement of "ApoE-reconstituted lipoprotein could overcome BBB, BBTB and enormously accumulated at the glioma site where nutrient is lack due to tumor growth".

Response: Good points. We agree with the reviewer that the stability of CaP-LNC might also contribute to the significant low uptake of CaP-LNC in tumor compared with CaP-rHDL, and the serum stability and pharmacokinetic study should be conducted to guide *in vivo* data analysis. Therefore, we firstly studied the serum stability of siRNA loaded in CaP-LNC and CaP-rHDL by incubating the siRNA-loaded CaP-LNC and CaP-rHDL in 10% serum at 37°C for different time durations and examining the stability of siRNA *via* agarose gel

electrophoresis thereafter. As shown in Fig. 1d, NC-siRNA carried by CaP-LNC and CaP-rHDL were both stable up to 8 h, while naked NC-siRNA degraded after only 2-h incubation, confirming that both CaP-LNC and CaP-rHDL can shield siRNA from degradation under physiological condition. Secondly, according to the reviewer's suggestion, we constructed an alternate control nanoparticle according to the formulation described in Ref 19 by shielding the surface of CaP-LNC with PEG-lipid (supplementary Table 1 and supplementary Fig. 6), and compared the *in vivo* distribution profile of this control nanoparticle with that of CaP-rHDL in NOD/SCID mice model bearing glioblastoma derived from patient GICs using Cy5-siRNA as the cargo and indicator. It was found that, consistent with CaP-LNC, this PEGylated CaP-LNC (Cy5-CaP-LNC-PEG) showed poor capacity in tumor targeting and permeation while Cy5-siRNA carried by CaP-rHDL accumulated and penetrated into the tumor site much more efficiently (Fig. 6f-h), suggesting that the incorporation of ApoE in the nanocarrier did play an important role in enhancing its glioblastoma-targeting efficiency. According to the reviewer's suggestion, these additional data were incorporated in the revised manuscript, please see Fig. 1d, Fig. 6f-h, supplementary Table 1 and supplementary Fig. 6, and the discussion about how CaP-rHDL achieved higher tumor targeting efficiency was carefully addressed, please see Line 373-382.

2. A dual dye labeled CaP-rHDL (siRNA labeled with FAM, particle's membrane

labeled with DiI) was used to investigate the de-assemble ability of nanocarrier. As micropinocytosis is an entire particle internalization pathway, a time-dependent imaging of dual dyes in subcellular components is suggested to track the disassembling of nanoparticles.

Response: Good suggestion. Accordingly, a time-dependent imaging of the dual dyes in subcellular components was performed to track the disassociation of the nanocarrier. As shown in Fig. 7b, the dual dye labeled CaP-rHDL did exhibit a time-dependent disassembling process. In detail, after incubated with C6 cells for 2 h, the nanoparticles remained relatively intact with FAM-labeled siRNA (green) found well colocalize with the DiI-embedded lipid membrane (red). With the increase of incubation time, the extent of colocalization decreased and the colocalization coefficient of FAM-siRNA to DiI decreased from 0.71 at 2 h to 0.46 at 12 h, suggesting that siRNA-loaded CaP-rHDL did self-disassemble following cellular internalization. After incubated for 12 h, in certain cells, the green fluorescent of FAM-siRNA was found well diffuse throughout the cytoplasm. Collectively, the time-dependent imaging study suggested the high efficiency of CaP-rHDL in cytosol siRNA release. Related data and discussion please see Fig. 7b and Line 426-433.

3. The nanoparticle name is confusing. Authors are quite liberal with the use of term, CaP-rHDL, where it stands for siRNA loaded nanoparticle in Figure 1a but also was used for imaging studies without siRNA loading. Yet, another different name was used

for the toxicity study (Fig.10) (ATF5 siRNA-loaded CaP-rHDL). Please be consistent.

Response: Thanks for the comment. We had renamed the different nanoparticles accordingly. CaP-rHDL was used as the general abbreviation of the nanocarrier; DiI, DiI-labeled CaP-rHDL was abbreviated as DiI-CaP-rHDL and DiR-CaP-rHDL, respectively; FAM-siRNA, Cy5-siRNA-loaded CaP-rHDL as FAM-CaP-rHDL and Cy5-CaP-rHDL, respectively; PEGylated CaP-LNC loaded with Cy5-siRNA as Cy5-CaP-LNC-PEG; NC-siRNA and ATF5-siRNA-loaded CaP-rHDL as NC-CaP-rHDL and ATF5-CaP-rHDL, respectively. Revision has been made accordingly in article; please see Line 182-183, 387-388, 140-141, 376, 370, 109 and 103 in the revised manuscript.

4. The nanoparticles labeled with DiI or DiR on the particle membrane were used for examining the delivery mechanism by fluorescence imaging technology. It is better to label the siRNA payload with fluorescence dyes, such as FAM or NiR dye, to directly monitor siRNA delivery in vitro or in vivo.

Response: Good suggestion. DiI and DiR, long-chain dialkylcarbocyanine lipophilic tracers widely used for liposome, cell plasma membrane and reconstituted high density lipoprotein labeling that generally not transfer from the labeled to unlabeled ones¹⁻⁵, were utilized for examining the delivery mechanism of the carrier (Fig. 2-5, Fig. 6a-e and Fig. 7). According to the reviewer's suggestion, FAM and Cy5 were also used to label the siRNA payload to directly monitor siRNA delivery *in vitro* (Fig. 7) and *in vivo* (Fig. 6f-h),

respectively.

5. Please correct some inconsistent dye labeling in Fig 6 (DiR-labeled) vs in the text (DiI-labeled). The explanation for switching DiI labeling (*in vitro* study) to DiR labeling (*in vivo* study) should also be given.

Response: Thanks for the suggestion and sorry for the confusion. In our study, DiI-labeling was used for both *in vitro* cellular uptake study (Fig. 2-5 and Fig. 7) and *in vivo* distribution analysis in frozen brain slides (Fig. 6e). In contrast, DiR labeling was used for *in vivo* real-time fluorescent imaging (Fig. 6a-d). DiI and DiR were both amphiphilic fluorescent probes that can be incorporated into the membrane of CaP-LNC and CaP-rHDL for fluorescent labeling. For confocal microscopy analysis, DiI is one of the most common dyes for staining artificial and biological membranes because it has extremely high extinction coefficient and short excited-state lifetimes (~1 nanosecond) in lipid environments, and it can be excited by a 561-nm laser line to yield emission in the 575- to 632-nm range (data according to Invitrogen; Carlsbad, CA), giving good signals under confocal laser scanning microscope. In contrast, for *in vivo* real-time fluorescent imaging, DiR (ex. 748 nm, em. 782 nm) labeling was used instead of DiI labeling because the longer excitation and emission wave length of DiR can minimize the animals' background autofluorescence and enable stronger tissue penetration. Clarification and the explanation for switching DiI labeling to DiR labeling was incorporated in the revised manuscript, please see Line 345-349.

6. *Some important references are missed in the manuscript including the first lipoprotein redirecting report (PNAS, 2005, 102, 17757-17762) as well as the reference to the selected ATF5-siRNA sequences.*

Response: Thank you for the suggestion. We have carefully read the publication suggested (PNAS, 2005, 102, 17757-17762), finding it interesting and inspirational. Therefore, we cited it in our revised manuscript, please see Reference 52. The ATF5-siRNA sequences were designed and provided by Genepharma Co. (Shanghai, China) and Ribobio. (Guangzhou, China). The gene knockdown efficiency of the siRNA sequences was confirmed by qPCR and western blot. Related data please see Fig. 8a, b, and Fig. 9b.

7. *Please define abbreviations when they appear in the manuscript at the first time, such as Ras-GTP, DDS, CCK-8, etc.*

Response: Thanks for the suggestion. We have defined the abbreviations when they appear in the manuscript at the first time. Please see Line 63, 589 and 456-457.

Reviewer: 2

Comments:

Brain tumor delivery is a significant concern. The studies involve a potential improvement in the specificity of delivery based on a system specifically geared towards ras activated cells. The initial phases of the development are interesting, but the biologic aspects of the studies are relatively weak and would benefit from substantial additional effort. In addition, the data are poorly presented with substantial data potentially better presented as data not shown or in the supplemental data.

1. The authors make assumptions regarding ras activity based on shRNA against three Ras family members and immunoblots. This is inadequate. The measurement of ras activity, not ras protein expression, should be performed with every model.

Response: Very good suggestion. Accordingly, using G-LISA Ras activation assay, we measured the Ras activity in C6 glioblastoma cells, patient-derived GICs, human glioblastoma cell lines (U87 and U251 cells), pancreatic cancer cells (MIA PaCa-2 and BxPC-3 cells) and colorectal cell lines (SW-480 and Caco-2 cells) before and after shRNA-mediated Ras knockdown (Fig. 4-5 and supplementary Fig. 2-3), and found a nice linear relationship between cellular Ras activity and the efficiency of cellular uptake of CaP-rHDL. Please check the detailed data in Fig. 4-5 and supplementary Fig. 2-3, Line 254-276 and Line 325-329 in the revised manuscript.

2. The authors have performed almost all studies in C6 glioma. The rationale for this model is lacking. It only makes sense to us C6 glioma if a rat host is to be used. The use of nude athymic mice supports a more relevant focus on the human models. The models presented are poorly characterized, with no molecular information. This manuscript would benefit from a much greater number of models representing a spectrum of molecular groups, at least *in vitro*.

Response: Great points. We agree with the reviewer that more models will make the study more reliable. C6 cell, as a classical rat glioma cell line with close relation with RAS pathway for cell proliferation⁶, was used as one of the key cell models. Nude athymic mice were used for the establishment of the animal model bearing C6 glioblastoma (Line 714-716). According to the reviewer's suggestion, here we further used human glioblastoma cell lines (U87 and U251 cells), pancreatic cancer cells (MIA PaCa-2 and BxPC-3 cells) and colorectal cell lines (SW480 and Caco-2 cells) as models to study the mechanism of CaP-rHDL internalization *in vitro*. Ras protein level and Ras activity in these cells were determined before and after shRNA-mediated Ras knockdown, and the efficiency of cellular uptake of CaP-rHDL was found nicely linear with the cellular Ras activity (Fig. 4 and supplementary Fig. 2-3).

To determine whether the cancer cell-targeting delivery strategy can be translated into a human system, we further investigated the cellular uptake of CaP-rHDL and its macropinocytosis dependence in a patient-derived glioblastoma model. As glioblastoma is composed of various subclones of

heterogeneous cells, we utilized patient-derived glioblastoma initiating cells (GICs) (supplementary Fig. 4) which closely mirror the phenotype and genotype of primary tumors for the evaluation. In terms of the molecular information about GICs, in addition to the measurement of Ras expression and Ras activity (Fig. 5), the expression of the general stem cell markers including SOX2, CD133 and Nestin were also detected (supplementary Fig. 4a). Limiting dilution study was also performed to evaluate the single cell cloning ability of GICs (supplementary Figure 4b, c). NOD/SCID mice model bearing glioblastoma derived from patient GICs was also applied to determine the *in vivo* pharmacokinetic and pharmacological effect of the CaP-rHDL nanoformulation (Fig. 6f-h, Fig. 8e-f, Fig. 9b-c and Fig. 9e), which nicely support our hypothesis that CaP-rHDL not only efficiently delivers siRNA to the target tissue, but also facilitates the gene knockdown activity to exert its antitumor efficacy. The related data have been added in the revised manuscript, please see Fig. 4-5, Fig. 6f-h, Fig. 8e-f, Fig. 9b-c, Fig. 9e and supplementary Fig. 2-4.

3. The glioma initiating cells (GICs) lack any kind of validation. To claim GICs, it is essential to perform in vivo and in vitro analysis. The optimal approach would involve the separation of stem-like and differentiated progeny and tumor initiation in vivo with in vitro limiting dilution studies.

Response: Good suggestion. According to the reviewer's suggestion, data about GICs validation was provided in supplementary Fig. 4. The expression of the

general stem cell markers including SOX2, CD133 and Nestin were observed in GICs (supplementary Fig. 4a). Limiting dilution study was also performed to evaluate the single cell cloning ability of GICs derived from patients after 10 passages, four spheres were formed in two 96-well plates, and the self-renew and differentiation ability of the subclones were shown in supplementary Fig. 4b-e. In addition, the mice model bearing patient-derived intracranial glioblastoma was established with 1000 GICs spheres. Related information please see supplementary Fig. 4 and Line 305-308 and Line 716-721 in the revised manuscript.

4. The shRNA studies for each Ras family member is somewhat concerning. Having the same response for all the Ras members suggests a lack of specificity in the effects. It would be useful to document the Ras activity after shRNA treatment. Similar studies should be performed in normal cells.

Response: Thanks for the suggestion. Indeed, the same response for all the Ras members did suggest a lack of specificity in CaP-rHDL uptake among the different Ras members. Since macropinocytosis is a general process in Ras-activated cancer cells⁷⁻¹⁰, it is not surprising to see that knockdown of either Ras member expression uniformly reduces macropinocytosis-dependent cellular uptake of CaP-rHDL. As all the three Ras proteins KRas, HRas and NRas expressed in mammalian cells can promote oncogenesis when mutation occurs or overexpresses, which produces functional redundancy of GTPase activity and

activates downstream micropinocytosis pathway¹¹, we believe this macropinocytosis-mediated cellular uptake of CaP-rHDL would work despite the activation of the different Ras members and serve as a universal nanocarrier for targeting therapy of Ras-activated cancer cells.

According to the reviewer's suggestion, using G-LISA Ras activation assay, we measured the Ras activity in C6 glioblastoma cells, patient-derived glioblastoma initiating cells, human glioblastoma cell lines (U87 and U251 cells), pancreatic cancer cells (MIA PaCa-2 and BxPC-3 cells) and colorectal cell lines (SW-480 and Caco-2 cells) before and after shRNA-mediated Ras knockdown (Fig. 4-5 and supplementary Fig. 2-3), and found a nice linear relationship between cellular Ras activity and the efficiency of cellular uptake of CaP-rHDL. Please check the detailed data in Fig. 4-5, supplementary Fig. 2-3, Line 254-276 and Line 325-329 in the revised manuscript.

5. The in vivo outcome is interesting but has only a single, suboptimal model with a single replicate. The preclinical efficacy is the most important issue. There should be a greater effort to show that these results can be generalized. Also, the tumors should be analyzed over the long term to demonstrate delivery of the siRNA and effects on the cells. The data in Figure 9 is biased in favor of response because it is too early and intensive.

Response: Thanks for the suggestion. In order to see if the results can be generalized, additional evaluation was performed on animal model established

with patient-derived GICs, and the results were shown in Fig. 9 and supplementary Fig. 10b. Similarly, ATF5-CaP-rHDL efficiently induced apoptosis at the glioma site in mice bearing intracranial patient-derived glioblastoma (Fig. 9c). In addition, the long term effect of ATF5-CaP-rHDL treatment was reflected by the survival in both animal models following four or five injections (Fig. 9d-e).

6. The sphere responses are not properly designed. The images are not useful. The proper studies are in vitro limiting dilution studies with single cells.

Response: Thanks for the suggestion. In the revised manuscript, the *in vitro* anti-glioma activity of ATF5-CaP-rHDL was showed in Fig. 8 with data about inhibition of the growth of the C6 glioblastoma spheroids deleted. In contrast, the anti-glioblastoma activity of ATF5-CaP-rHDL on patient-derived GICs was demonstrated in Fig. 8e-f.

7. The toxicity data are not well designed. First, the single astrocyte line is not well characterized and the inclusion of these cells in a greater number of experiments for both entry and toxicity would be important. Second, in the in vivo studies, the imaging suggests that delivery to some organs may be greater than the tumor. The weight analysis and organ analysis is performed at a very early time point. The longer-term effects should be considered.

Response: Good suggestions. First, according to the reviewer's suggestion, the

astrocyte is characterized by GFAP staining (supplementary Fig. 1). The efficiency of cellular uptake of CaP-rHDL in astrocytes was shown in Fig. 2d and the toxicity of the designed nanoformulation on astrocytes was shown in supplementary Fig. 7. Second, indeed, the current nanocarrier containing ApoE as the functional moiety exhibited relatively high accumulation in the liver. In our future work, ApoE will be replaced with other proteins or peptides can also be specifically internalized by macropinocytosis but with lower affinity to the non-targeted organs. In this revised manuscript, the long term toxicity of ATF5-CaP-rHDL was evaluated in normal mice following injection every two days for eight times. After the treatment, blood chemistry test and morphological observation were performed with no significant changes detected in all the organs including kidney and liver where the nanoparticles exhibited high accumulation, suggesting the safety of ATF5-CaP-rHDL formulation. The related results were shown in Fig. 10 and Line 553-557 in the revised manuscript.

8. The ability to deliver across the blood-brain barrier was not tested and the targeting of the reagents are exciting but the entire evaluation is not optimal and are not at the level for a high impact journal, yet.

Response: Thanks for the suggestions. Additional experiments have been performed to test the ability of CaP-rHDL to penetrate across the blood-brain barrier (BBB) with blood vessels visualized by anti-CD31. As shown in the supplementary Fig. 5, CaP-rHDL was found efficiently penetrate through the

BBB in both glioblastoma-bearing animal models.

According to the reviewer's suggestion, in this revised manuscript, additional cell models and animal model bearing patient-derived glioblastoma were further used for studying the mechanism and evaluating the efficacy of CaP-rHDL-mediated glioblastoma-targeting siRNA delivery (Fig. 4, 5, 6, 8 and 9). Hope all our revisions and responses will be satisfactory and this revised version will be acceptable for publication in *Nature Communications*.

Reviewer: 3

Summary:

In this manuscript the authors provide evidence that ApoE-rHDL nanoparticles containing a calcium phosphate (CaP) core can be taken up effectively by glioma cells in culture (monolayer and spheroid) and in intracranial glioma in mice. Furthermore, they demonstrate that such nanoparticles can be employed to deliver siRNA targeting a pro-survival protein, ATF5, thereby inhibiting intracranial tumor growth and prolonging survival of tumor-bearing mice. The authors claim two other novel aspects of their work: 1) The mechanism of uptake of the CaP-rHDL nanoparticles entails their incorporation into macropinosomes. 2) Macropinocytosis is Ras-dependent, suggesting that the NPs may be particularly useful for targeting "Ras-activated glioblastoma".

General Comments:

While the use of rHDL nanoparticles for delivery of therapeutic cargo is not entirely new, their successful use for targeted siRNA delivery to glioblastoma is potentially very important. Overall the aspects of the current studies demonstrating selective uptake of the CaP-rHDL particles and effective ATF5 suppression by siRNA in vitro are well designed and executed. The evidence that macropinocytosis plays a key role in the process is also convincing. The endogenous Ras and Rac pathways are known to play an essential role in the control of macropinocytosis in both normal and transformed cells, so it is not surprising that knockdown of Ras expression results in reduced uptake of CaP-rHDL nanoparticles. However, the claim that that such nanoparticles represent a specific therapeutic approach for selective targeting of tumors driven by activated Ras is confusing, and the studies fall short of demonstrating this. The problem is, in part, related to the authors' use of the terms "Ras-activated" glioblastoma and "Ras-transformed cell line". Activating mutations in Ras are rare in human GBM, and the majority of the studies in this report were performed with the C6 rat glioma cell line, which does not harbor Ras mutations and thus is not a "Ras-activated cell line". It is true that expression of endogenous Ras protein is relatively high in these cells compared with normal astrocytes but this is not the same as a "Ras-activated" tumor where Ras mutation (e.g., V12) results in constitutive activation of Ras pathways and is a driver of tumorigenesis and tumor progression.

Response: Great comments. We agree with the reviewer that activating mutations in Ras are rare in human GBM. According to the previous reports, the RAS/RAF pathway activation in glioblastoma is achieved much more frequently by copy number gains of RAS/RAF and/or upstream growth factor (receptor) than by activating RAS/RAF mutations^{12,13}. Simultaneously, the level of Ras-GTP in glioma cell lines and glioma specimens is markedly elevated compared to the normal cells and brain specimens, and is comparable to that in oncogenic Ras transformed fibroblasts¹⁴⁻¹⁷. As expected, C6 cell, a classical rat glioma cell line, showed a close relation between RAS activity and cell proliferation ability⁶. In this revised work, beside C6 cells, human GBM cell lines U87, U251 and patient-derived GICs, other cancer cells such as human pancreatic cancer line MIA PaCa-2, human colon cancer cells SW480 with high level of Ras mutation were also used as the cell models. G-LISA Ras activation assay showed that the glioblastoma cells exhibited comparable high Ras activity as that of those cancer cell lines with high Ras mutation (Fig. 4b), confirming the high Ras activity in glioblastoma cells. Therefore, both literatures and our data suggested that although activating mutations in Ras are rare in human glioblastoma, glioblastoma cells remain to be Ras-activated cells.

The reviewer is right that our previous manuscript fall short of demonstrating that the selective targeting of CaP-rHDL to tumors is driven by activated Ras. To address this point, we further used human glioblastoma cell lines, human pancreatic cancer cells and colorectal cell lines as cell models with different Ras

activity to test the CaP-rHDL internalization mechanism *in vitro* and witnessed a nice linear relationship between cellular Ras activity and the efficiency of cellular uptake of CaP-rHDL (Fig. 4-5). In addition, reducing Ras activity by knocking down Ras expression also significantly inhibited the macropinocytosis-dependent cellular uptake of CaP-rHDL (Fig. 4-5 and supplementary Fig. 2-3). These data collectively suggested that the selective targeting of CaP-rHDL to tumors is driven by activated Ras. Additional data and discussion please see Fig. 4-5, supplementary Fig. 2-3, Line 254-276 and Line 325-329.

Specific Comments:

1) Measurements of total endogenous Ras protein expression levels (Fig. 5a) do not necessarily indicate that Ras is "hyperactivated". Determination of the percentage of activated Ras (e.g., by G-LISA Ras activation assays) would be required to draw this conclusion.

Response: Very good suggestions. Accordingly, G-LISA Ras activation assay was performed to determine Ras activity in different cell lines before and after shRNA-mediated Ras knockdown (Fig. 4-5 and supplementary Fig. 2-3). It was found that the glioblastoma cells exhibited comparable high Ras activity as that of those cancer cell lines with high Ras mutation, and those cells with higher Ras activity such as MIA PaCa-2, SW480 showed much higher cellular association of DiI-labeled CaP-rHDL, compared with their control cells with lower Ras activity,

BxPC-3 and Caco-2, respectively. A straightforward linear relation was achieved between the cellular uptake of CaP-rHDL and the intracellular Ras-GTP level ($R^2=0.8687$) (Fig. 4c and Fig. 5e) Related data has been incorporated in the revised manuscript, please see Fig. 4-5, supplementary Fig. 2-3, Line 254-276 and Line 325-329.

2) The conclusion that CaP-rHDL nanoparticles "penetrated deep and distributed extensive(ly) in the C6 glioblastoma spheroids" (pg 16) seems inconsistent with the confocal images in Supplementary Fig. 2, which show most of the DiI-labeled CaP-rHDL concentrated around the periphery of the spheroids. Regarding the same point, in Fig. 5b, the images of patient-derived spheroids appear to be whole mounts, so it is not clear whether the fluorescent cells are mostly on the surface or the interior.

Response: Good points. We agree with the reviewer that our previous conclusion that CaP-rHDL nanoparticles "penetrated deep and distributed extensive in the C6 glioblastoma spheroids" seems inconsistent with the confocal images in Supplementary Fig. 2. Indeed, when the tumor spheroids grow up to a certain extent, necrotic cores occur at the center of the spheroids. This could be the major reason why most of the DiI-labeled NP concentrated around the periphery of the spheroids as demonstrated previously. Reviewer 2 also points out that this 3D C6 glioblastoma spheroid is not properly designed. Therefore, we deleted the related data derived from C6 glioblastoma spheroids. In this revised manuscript,

the high permeability of CaP-rHDL was reflected by its deep penetration and extensive distribution at the glioblastoma sites in the brain of the tumor-bearing animals (Fig. 6 and Supplementary Fig. 5). In the case of patient-derived GICs spheroids, where less necrotic cores occur, as shown in Fig. 5b and supplementary video 1-2, deep penetration of CaP-rHDL within the spheroids was witnessed.

3) The small sizes of some of the illustrations, particularly in the complex figures, make it impossible to see if the conclusions are supported. For example, in Fig. 8d, the authors conclude that treatment with ATF5 siRNA "led to hyperchromatism of the cell nuclei". That panel should be moved to the supplementary data and expanded so that nuclear morphology can be easily assessed.

Response: Thank you for the suggestion. We have moved the panel to the supplementary data (supplementary Fig. 8) accordingly and the hyperchromatism of the cell nuclei was indicated by arrowhead.

4) In Fig. 8e, we are shown one spheroid for each condition evidence that anti-ATF5-CaP-rHDL nanoparticles inhibit growth/viability of C6 spheroids. This conclusion needs to be supported by quantitative data such as average spheroid diameter or volume, with an appropriate and statistical analysis.

Response: Thanks for the suggestion. As discussed above, the 3D C6 glioblastoma

spheroid model is not properly designed, we deleted the related data. Instead, the anti-glioma activity of ATF5-CaP-rHDL was evaluated by anti-proliferation and cell apoptosis assays with quantitative data provided and an appropriate and statistical analysis performed. As shown in Fig. 8 and supplementary Fig. 9, ATF5-CaP-rHDL exhibited much higher tumor-cellular toxicity and superiority in inducing cell apoptosis with the percentage of early apoptosis in cells treated with ATF5-CaP-rHDL ($53.2\% \pm 10.5\%$) much higher than that of those treated with ATF5-CaP-LNC ($11.1\% \pm 1.3\%$) and NC-CaP-rHDL ($7.1\% \pm 0.15\%$). The related results were shown in Fig. 8, supplementary Fig. 9 and Line 459-471.

5) The argument that the increased apoptosis (Fig. 9) and the improved survival (Fig. 10) of mice treated with anti-ATF5 nanoparticles is specifically related to suppression of ATF5 would be strengthened by IF or WB results demonstrating that ATF5 is suppressed in the tumor tissue.

Response: Thanks for the suggestion. Accordingly, we examined the level of ATF5 at the tumor site by western blot, finding that compared with saline, ATF5-CaP-rHDL suppressed ATF5 expression by 80% (Fig. 9b). Such description was also added in the revised manuscript, please see Fig. 9b and Line 501-503.

6) The paper contains numerous grammatical and typographical errors throughout.

Response: Thanks for the suggestion. The grammatical and typographical errors

have been corrected. We hope this revised version could be much more fluent and neat.

References:

1. Costales, P. *et al.* K domain CR9 of low density lipoprotein (LDL) receptor-related protein 1 (LRP1) is critical for aggregated LDL-induced foam cell formation from human vascular smooth muscle cells. *J. Biol. Chem.* **290**, 14852-14865 (2015).
2. Yu, Y. L. *et al.* Comparison of commonly used retrograde tracers in rat spinal motor neurons. *Neural Regen. Res.* **10**, 1700-1705 (2015).
3. Kim, S. H. *et al.* Targeted intracellular delivery of resveratrol to glioblastoma cells using apolipoprotein E-containing reconstituted HDL as a nanovehicle. *Plos One* **10**, e135130 (2015).
4. Chen, J. *et al.* Ligand conjugated low-density lipoprotein nanoparticles for enhanced optical cancer imaging *in vivo*. *J. Am. Chem. Soc.* **129**, 5798-5799 (2007).
5. Kalchenko, V. *et al.* Use of lipophilic near-infrared dye in whole-body optical imaging of hematopoietic cell homing. *J. Biomed. Opt.* **11**, 50507 (2006).
6. Aoki, K., Yokosuka, A., Mimaki, Y., Fukunaga, K. & Yamakuni, T. Nobiletin induces inhibitions of Ras activity and mitogen-activated protein kinase kinase/extracellular signal-regulated kinase signaling to suppress cell

- proliferation in C6 rat glioma cells. *Biol. Pharm. Bull* **36**, 540-547 (2013).
7. Kamphorst, J. J. *et al.* Hypoxic and Ras-transformed cells support growth by scavenging unsaturated fatty acids from lysophospholipids. *Proc. Natl. Acad. Sci. U. S. A.* **110**, 8882-8887 (2013).
 8. Commisso, C. *et al.* Macropinocytosis of protein is an amino acid supply route in Ras-transformed cells. *Nature* **497**, 633-637 (2013).
 9. Alonso-Curbelo, D. *et al.* RAB7 counteracts PI3K-driven macropinocytosis activated at early stages of melanoma development. *Oncotarget* **6**, 11848-11862 (2015).
 10. Bar-Sagi, D. & Feramisco, J. R. Induction of membrane ruffling and fluid-phase pinocytosis in quiescent fibroblasts by ras proteins. *Science* **233**, 1061-1068 (1986).
 11. Prior, I. A., Lewis, P. D. & Mattos, C. A comprehensive survey of Ras mutations in cancer. *Cancer Res.* **72**, 2457-2467 (2012).
 12. Jeuken, J., van den Broecke, C., Gijzen, S., Boots-Sprenger, S. & Wesseling, P. RAS/RAF pathway activation in gliomas: the result of copy number gains rather than activating mutations. *Acta. Neuropathol.* **114**, 121-133 (2007).
 13. Lo, H. W. Targeting Ras-RAF-ERK and its interactive pathways as a novel therapy for malignant gliomas. *Curr. Cancer Drug Targets* **10**, 840-848 (2010).
 14. Guha, A., Feldkamp, M. M., Lau, N., Boss, G. & Pawson, A. Proliferation of human malignant astrocytomas is dependent on Ras activation. *Oncogene* **15**, 2755-2765 (1997).

15. Guha, A. Ras activation in astrocytomas and neurofibromas. *Can J. Neurol Sci.* **25**, 267-281 (1998).
16. Feldkamp, M. M., Lala, P., Lau, N., Roncari, L. & Guha, A. Expression of activated epidermal growth factor receptors, Ras-guanosine triphosphate, and mitogen-activated protein kinase in human glioblastoma multiforme specimens. *Neurosurgery* **45**, 1442-1453 (1999).
17. Feldkamp, M. M., Lau, N., Roncari, L. & Guha, A. Isotype-specific Ras.GTP-levels predict the efficacy of farnesyl transferase inhibitors against human astrocytomas regardless of Ras mutational status. *Cancer Res.* **61**, 4425-4431 (2001).

REVIEWERS' COMMENTS:

Reviewer #1 (Remarks to the Author):

The authors have addressed all comments. Great work.

Reviewer #3 (Remarks to the Author):

The authors have done a good job addressing the concerns I raised in my original comments.

My only remaining minor concern relates to Supplementary Fig. 8, where it is still very difficult to see that the nuclei pointed out by the arrows are really hyperchromatic. This seems to be a problem with the blue color, which does not show up well, plus the low magnification and the small size of the panels. Perhaps the authors could try showing an inset at higher magnification?

Below is a detailed description of our revision according to the review comments.

Journal: *Nature Communications* (NCOMMS-16-01632A-Z)

Title: “Lipoprotein-Biomimetic Nanostructure Enables Efficient Targeting Delivery of siRNA to Ras-Activated Glioblastoma Cells *via* Macropinocytosis”

Review comments:

Reviewer: 3

Minor Comments:

1) My only remaining minor concern relates to Supplementary Fig. 8, where it is still very difficult to see that the nuclei pointed out by the arrows are really hyperchromatic. This seems to be a problem with the blue color, which does not show up well, plus the low magnification and the small size of the panels. Perhaps the authors could try showing an inset at higher magnification?

Response: Thank you for the suggestion. An inset at higher magnification with the hyperchromatic cell nuclei indicated by arrowhead has been added, please see Supplementary Fig.8.